# Diffusion Generative Modeling for Spatially Resolved Gene Expression Inference from Histology Images

**Sichen Zhu**[*], **Yuchen Zhu**[*], **Molei Tao**[†], **Peng Qiu**[†]
Georgia Institute of Technology
{sichenzhu, yzhu738, mtao}@gatech.edu, peng.qiu@bme.gatech.edu

## Abstract

Spatial Transcriptomics (ST) allows a high-resolution measurement of RNA sequence abundance by systematically connecting cell morphology depicted in Hematoxylin and Eosin (H&E) stained histology images to spatially resolved gene expressions. ST is a time-consuming, expensive yet powerful experimental technique that provides new opportunities to understand cancer mechanisms at a fine-grained molecular level, which is critical for uncovering new approaches for disease diagnosis and treatments. Here, we present **Stem** (**S**pa**T**ially resolved gene **E**xpression inference with diffusion **M**odel), a novel computational tool that leverages a conditional diffusion generative model to enable in silico gene expression inference from H&E stained images. Through better capturing the inherent stochasticity and heterogeneity in ST data, **Stem** achieves state-of-the-art performance on spatial gene expression prediction and generates biologically meaningful gene profiles for new H&E stained images at test time. We evaluate the proposed algorithm on datasets with various tissue sources and sequencing platforms, where it demonstrates clear improvement over existing approaches. **Stem** generates high-fidelity gene expression predictions that share similar gene variation levels as ground truth data, suggesting that our method preserves the underlying biological heterogeneity. Our proposed pipeline opens up the possibility of analyzing existing, easily accessible H&E stained histology images from a genomics point of view without physically performing gene expression profiling and empowers potential biological discovery from H&E stained histology images. Code is available at: https://github.com/SichenZhu/Stem.

## 1 Introduction

Histology imaging of Hematoxylin and Eosin (H&E) stained tissues has been an important, longstanding tool in biomedical research and clinical diagnosis. H&E stained histology images provide rich information about the tissue composition and cell morphology at a cellular, microscopic level. In recent years, the emergence of Spatial Transcriptomics (ST) technology has provided an opportunity to deepen our understanding of these H&E images and tissue slides to a more fine-grained molecular level. ST technology segments centimeter-size Whole Slide Images (WSIs) into hundreds of spots with a micrometer-size diameter and generates gene expression profiling of the tissue within each spot (Ståhl et al., 2016). ST has seen prominent applications in biomedical and clinical scenarios. By connecting genomics information to cells' spatial location within the tissue, ST captures the underlying biological heterogeneity across various cells in different locations and reveals the cancer microenvironment for better targets in treatment (Lewis et al., 2021; Williams et al., 2022).

While being a promising tool to explore potential relationships between cell morphology and gene expression patterns, existing ST technology such as Visium (Ståhl et al., 2016) are less accessible due to its substantial cost in time and experimental preparation work in wet labs. On the other hand, H&E stained images are enriched in clinical settings due to their low cost and wide application.

---

[*]Equal contribution.
[†]Corresponding authors.

To leverage the abundant, easily accessible histology images and overcome the currently limited accessibility of ST technology, one natural idea is to computationally infer gene expression profiles from H&E images to explore subcellular information implicitly contained within H&E images. Therefore, in the sequel, we try to ask and better address the following question:

*Can we develop a machine learning tool to computationally infer spatially resolved gene expression solely based on histology images?*

Several previous works have attempted to tackle this challenge with various approaches. These proposed methods approach the question from different perspectives, with some using only local spot image patch to infer gene expressions (Xie et al., 2024; Yang et al., 2023; He et al., 2020; Min et al., 2024), some making use of the spatial dependencies between image patches to enhance the prediction (Zeng et al., 2022; Zhang et al., 2024; Pang et al., 2021), and some taking both the local and global image information into consideration for better prediction (Chung et al., 2024; Wang et al., 2024a). However, the central assumption in all these aforementioned works is that the gene expression prediction can be treated as a (nonlinear) regression task between the (partial or whole slide) histology image and the expression data, i.e., to find a deterministic prediction function that takes in the histology images and outputs the predicted gene expressions.

This thought, while seemingly sound at first sight, has several inherent limitations. First of all, existing methodologies often perform such regression by outputting averaged gene expressions of spots in the training dataset with similar images as the test histology image. This requires comparing the image embedding of the test histology image and all of the training datasets, suggesting that the computational resources required for model inference are proportional to the reference training dataset size. This scalability issue limits the utility of these approaches in realistic applications, where the reference dataset is at scale. Secondly, gene expression prediction as a regression task is intrinsically ill-posed. While the histology images contain a high level of information about the paired gene expression data, it is unlikely that this information renders an injective mapping between histology images and spatial transcriptomic data, not to mention the high level of noise and dropout contained in the gene expressions. Therefore, starting from the problem formulation phase, we must take the one-to-many scenario into consideration, i.e., similar histology images could correspond to distinct gene expression profiles due to biological heterogeneity. For example, even cells of the same cell type might be in different cell states or differ in their gene expressions due to their different spatial locations. Moreover, current model performance is often over-estimated due to an overly simplistic evaluation framework based primarily on Pearson correlation. Such a metric fails to reflect how well the model captures the biological and spatial heterogeneity within their prediction.

In response to these challenges and limitations, we present **Stem** (**S**pa**T**ially resolved gene **E**xpression inference with diffusion **M**odel), a novel framework for inferring spatially resolved gene expressions based on H&E stained histology images using conditional diffusion model. **Stem** tackles the question from a generative modeling perspective and learns a conditional distribution over the potentially associated gene expression profiles given the histology images, facilitating a one-to-many correspondence between the image and the transcriptomics data. **Stem** adopts the framework of the diffusion model (Ho et al., 2020; Song et al., 2020), which has showcased impressive capabilities in learning complex and multimodal conditional distribution across various domains (Rombach et al., 2022; Saharia et al., 2022). This strong power in conditional distribution modeling enables **Stem** to capture both similarity and heterogeneity across different genes and locations, resulting in higher prediction accuracy and robustness. **Stem** also leverages the recent great success of foundational models in computational pathology (Chen et al., 2024; Lu et al., 2024), which produces general-purpose embeddings for H&E stained histology images that implicitly contain information about the paired gene expression profiles of cells inside the images. **Stem** distills the image knowledge from these foundation models by using pooled embedding vector as the condition to represent histology images. This design saves the efforts of widely-used manual alignment between image and gene embedding that is widely adopted in existing methodologies (Xie et al., 2024; Chung et al., 2024; Yang et al., 2023). This design further reduces the computational cost in model training and inference and enables **Stem** to perform accurate and robust inference of spatially resolved gene expression profiles solely based on the image patch of a local spot. We evaluate **Stem** on four publicly available datasets from different tissue sources and sequencing platforms (kidney, Visium (Lake et al., 2023) & breast, SpatialTranscriptomics (Andersson et al., 2021) & prostate, Visium (Erickson et al., 2022) & mouse brain, Visium(Vicari et al., 2024)). Our proposed method demonstrates a re-

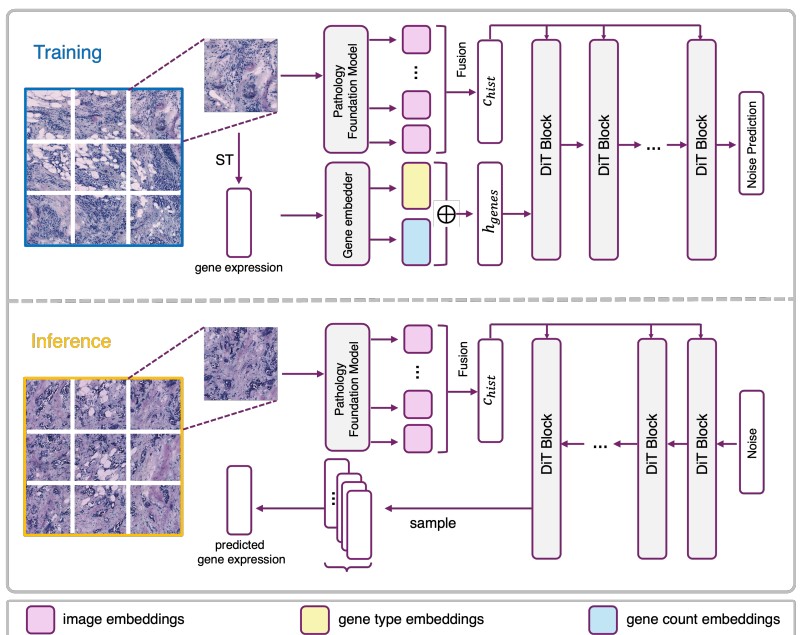

Figure 1: Overview of **Stem**. The input training data for **Stem** is ST datasets that contain both H&E images and spot-wise gene expression profiles. During training, gene counts and gene types are separately embedded and combined to serve as the input into DiT blocks. Images are cropped into $224 \times 224$ patches surrounding every spot and then tokenized via pathology foundation models. Fused image tokens serve as the conditions and are input into every DiT block. After training, gene expression output could be iteratively sampled conditioned on any input image patch.

markable state-of-the-art performance in the task of gene expression prediction and outperforms all existing approaches in terms of conventional evaluation metrics such as Mean Squared Error (MSE), Mean Absolute Error (MAE), and Pearson Correlation Coefficient (PCC). Through a simple study, we also show that PCC does not fairly evaluate the model's performance on this task since it ignores the spatial heterogeneity across different locations. We follow a similar setup as Xie et al. (2024) and define a new Relative Variation Distance (RVD) by comparing the generated prediction's relative and absolute gene variation with the ground truth data. A smaller gene variation distance suggests a better-preserved biological heterogeneity in the predictions similar to the original data, making it a better indicator for good predictions than PCC. Finally, we demonstrate that **Stem** produces biologically meaningful predictions by performing tissue structure annotations on unseen histology images with predicted gene profiles and compare with ground truth annotations provided by human pathologists. We also carry out a detailed ablation study on modules in the design space to ensure the algorithm's robustness.

To sum up, we have the following summarized contributions:

- We propose a novel algorithm **Stem** to predict spatially resolved gene expression profiles associated with H&E stained histology images using conditional diffusion model. To our best knowledge, this is the first generative modeling approach on this task.

- **Stem** integrates histology image information by leveraging pooled embedding from computational pathology foundation models and improves upon existing methodologies on required computational resources at test time.

- **Stem** achieves SOTA accuracy on multiple distinct datasets in terms of both standard metrics (MSE, MAE, PCC) and the newly proposed gene variation distance. **Stem** also succeeds in a difficult tissue structure annotation task by producing biologically meaningful gene expression predictions that are well aligned with the ground truth.

## 2 RELATED WORK

**Machine Learning Prediction of Gene Expression from Histology Images**   The task of predicting spatially resolved gene expression at a near single-cell resolution using patch-level H&E stained images has been approached as a regression task by several works. The seminal work of ST-Net (He et al., 2020) utilizes transfer learning to directly predict gene expression values from encoded histology images starting from a pre-trained model on ImageNet. HisToGene (Pang et al., 2021) and Hist2ST (Zeng et al., 2022) improve upon ST-Net by additionally introducing correlation among different patches and spatial location information into the model. Taking a step further towards including more information into the model, TRIPLEX (Chung et al., 2024) and M2ORT (Wang et al., 2024a) integrate nonlocal, holistic information of the histology image with the local information by extracting hierarchical, multi-resolution image features. In a different direction, BLEEP (Xie et al., 2024) and EGN (Yang et al., 2023) attempt to enhance prediction accuracy by retrieving gene expression values from the training set that are most similar to the test histology image query. BLEEP achieves this goal by aligning image and gene expression embedding through a CLIP-like contrastive learning loss (Radford et al., 2021) while EGN adopts the path of exemplar learning. We remark that while these mentioned approaches have shown promising performance on the gene expression prediction task, their idea has deep roots in the regression framework, which potentially hinders them from achieving more satisfactory results. Our proposed approach is inherently different from all the works mentioned above by taking a different route through generative modeling.

**Computational Pathology Foundation Models**   One of the central tasks in computational pathology is to obtain general-purpose embedding of histology image patches that can be used for downstream tasks such as gene expression prediction, cell phenotyping, and prognosis prediction (Jaume et al., 2024). Thanks to the abundance of histology images, powerful pathology foundation models have been trained on large-scale datasets (Ciga et al., 2022; Filiot et al., 2023; Vorontsov et al., 2023; Xu et al., 2024; Huang et al., 2023; Chen et al., 2024; Lu et al., 2024) with unimodal or multimodal self-supervised learning objectives, such as image-text contrastive learning (Radford et al., 2021), image captioning (Yu et al., 2022), DINO (Oquab et al., 2023), etc. Through massive pertaining, computational pathology foundation models implicitly learn to encode the tissue cell morphology into embeddings with rich information. Since cell morphology largely determines cell types, which further substantially affects the associated gene expression values, the embedding also partially encodes gene-related information that is highly beneficial for gene expression inference. In this work, we mainly leverage two foundational models, UNI (Chen et al., 2024) and CONCH (Lu et al., 2024) for extracting and aggregating histology image information. We also experiment with large-scale foundation models such as Virchow & Virchow-2 (Vorontsov et al., 2023; Zimmermann et al., 2024) and H-Optimus-0 (Saillard et al., 2024), which are trained with similar self-supervised methods as UNI but using a larger vision encoder.

**Diffusion Model for Multimodal data and Conditional Generation**   Diffusion models have shown remarkable performance in multimodal generation through the power of conditional generation, such as text-to-image (Rombach et al., 2022; Saharia et al., 2022; Esser et al., 2024), text-to-video (Singer et al., 2022; Ho et al., 2022), text-to-audio (Kreuk et al., 2022) and many more. One central element of conditional diffusion models is the conditioning mechanism since it affects how well information from different modalities fuses together. The flexibility in diffusion model design space allows the introduction of conditioned data modality into the model in various ways, where the cross-attention and modulation mechanisms are two mainstream approaches. For example, taking the literature of text-to-image diffusion model for demonstration, GLIDE (Nichol et al., 2021), Imagen (Saharia et al., 2022) and Stable-Diffusion (Rombach et al., 2022) take the route of cross attention and incorporates conditional information by attending the model to text condition embedding extracted with either learned or pre-trained encoders. In a different vein, PGv3 (Liu et al., 2024) recently introduced a new way to fuse information by performing joint attention between data and conditions using KV concatenation. In this work, we are also faced with the need for a conditioning mechanism to fuse the histology image information with the gene expression data. We select the modulation approach with adaptive LayerNorm, the same module used in DiT (Peebles & Xie, 2023), since it's parameter-efficient and fast for inference.

## 3 BACKGROUND

**Diffusion Model**  Diffusion models have shown tremendous success in generating complex data distribution, including numerous science applications such as protein design (Yim et al., 2023; Watson et al., 2023), quantum science (Zhu et al., 2024a;b), single cell analysis (Luo et al., 2024), chemistry (Duan et al., 2023) and neural science (Wang et al., 2024b). Before introducing our main algorithm and architecture, we first review some basics of the diffusion model in the setting of discrete time denoising diffusion (DDPMs) (Ho et al., 2020; Sohl-Dickstein et al., 2015).

Diffusion models are probabilistic models that are designed to learn data distribution $p_{\text{data}}(x)$ from samples. A diffusion model consists of two stochastic processes: forward noising and backward denoising processes. First, the forward process $q(x_t|x_0)$ is chosen to perturb the data distribution $p_{\text{data}}$ into a simple distribution $q_{\text{ref}}$. A common choice is to apply Gaussian noise to data $x_0$ gradually in $T$ steps and turns it into $p_T \approx q_{\text{ref}} = \mathcal{N}(0, \mathbf{I})$: $q(x_t|x_0) = \mathcal{N}(x_t; \sqrt{\bar{\alpha}_t} x_0, (1 - \bar{\alpha}_t)\mathbf{I})$, where $\bar{\alpha}_t$ are constant hyperparamters. With the parameterization trick, $x_t$ can be sampled by $x_t = \sqrt{\bar{\alpha}_t} x_0 + \sqrt{1 - \bar{\alpha}_t}\epsilon_t, \epsilon_t \sim \mathcal{N}(0, \mathbf{I})$.

The backward process reverts the forward noising process by iterative denoising $q_{\text{ref}}$ into $p_{\text{data}}(x)$. We parameterize the backward process as $p_\theta(x_{t-1}|x_t) = \mathcal{N}(\mu_\theta(x_t, t), \Sigma_\theta(x_t, t))$, where neural networks are used to predict the statistics of distribution from noisy data $x_t$. $p_\theta(x_{t-1}|x_t)$ is learned by maximizing the variational lower bound of the log-likelihood of true data $x_0$, which reduces to the minimization of following objective $\mathcal{L}(\theta) = \sum_t D_{\text{KL}}\big(q(x_{t-1}|x_t, x_0) \,\|\, p_\theta(x_{t-1}|x_t)\big)$. Once the diffusion model $p_\theta$ is well trained, new data can be sampled by simulating $x_T \sim \mathcal{N}(0, \mathbf{I})$ and sampling iteratively from $x_{t-1} \sim p_\theta(x_{t-1}|x_t)$ for $t = T, \ldots, 1$.

**Diffusion Conditional Generation**  Diffusion model is known to be a powerful conditional distribution learner (Rombach et al., 2022; Saharia et al., 2022; Peebles & Xie, 2023; Chen et al., 2023) and is capable of modeling distributions of the form $p_{\text{data}}(x|y)$, where $y$ is additional information such as class label, text, or histology images in our considered problem setting. This is enabled by allowing the neural network to take the condition $y$ as an additional input: $\epsilon_\theta(x_t, t, y)$. Notably, the condition $y$ often enters the model through its latent vector representation $h_y$. Therefore, diffusion models implicitly learn a mapping $h_y \rightarrow p_{\text{data}}(x|y)$ and often succeed in extrapolating to novel unseen conditions at inference time.

## 4 DIFFUSION GENERATIVE MODELING OF SPATIAL GENE EXPRESSION

**Problem Set-up**  Let $V \in \mathbb{R}^{L \times L \times 3}$ be a image patch of the H&E stained image, and $X \in \mathbb{R}^C$ be the associated gene expression profile, where $L$ is the image patch size and $C$ is the gene set size. We aim to infer $X$ when only the image patch $V$ is given. Existing methodologies treat the task as a regression problem and attempt to learn a deterministic function $f_{\text{expr}}$ such that $X \approx f_{\text{expr}}(V)$. However, while one histology image contains extensive information about the corresponding gene profile, it is unlikely to uniquely determine the gene expression vector due to tissue heterogeneity and uncertainty in the cellular microenvironment. This renders the mapping $f_{\text{expr}}$ between image patches $V$ and gene expressions $X$ non-injective. To address this potential issue, we treat the spatial gene expression prediction as a generative modeling task and aim to learn the conditional distribution of gene expression given the histology image $X \sim p_{\text{gene}}(X|V)$ from data sample pairs. This framework generalizes over the deterministic regression approach by potentially allowing one-to-many relationships between some image patches $V$ and gene expression $X$. Note that when the learned $p_{\text{gene}}(X|V)$ is a degenerate delta distribution, i.e., $p_{\text{gene}}(X|V) = \delta_{f_{\text{expr}}(V)}(x)$, we recover the deterministic regression setting. From now on, we focus on modeling the distribution of gene expression vectors conditioned on the associated histology image.

**Diffusion Generative Modeling of Gene Expressions**  We perform generative modeling of the conditional distribution $p_{\text{gene}}(X|V)$ with denoising diffusion model (DDPM). We choose the forward process $q(X_t|X_0, V) = \mathcal{N}(X_t; \sqrt{\bar{\alpha}_t} X_0, (1 - \bar{\alpha}_t)\mathbf{I})$, where $\bar{\alpha}_t$ are constant computed from the noise schedule hyperparameter $\beta_t$ as $\alpha_t = 1 - \beta_t, \bar{\alpha}_t = \prod_{s=1}^t \alpha_s$. We choose a linear noise schedule $\beta_t = \frac{t}{T}\beta_{\max} + (1 - \frac{t}{T})\beta_{\min}$. We write the backward process as $p_\theta(X_{t-1}|X_t, V) = \mathcal{N}(\mu_\theta(X_t, V, t), \sigma_t^2 \mathbf{I})$, where we fix the variance of the backward process to be untrained time de-

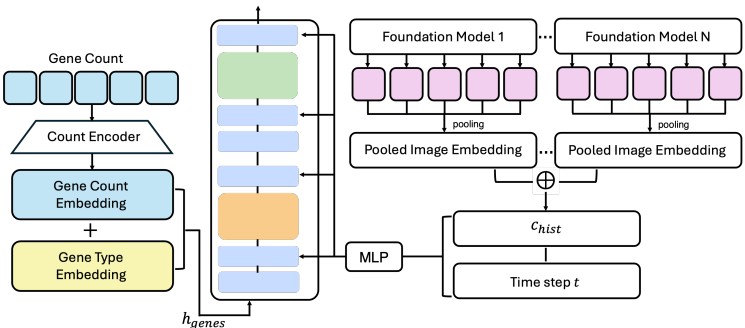

Figure 2: Visualization of neural network architecture in **Stem**. Histology Images are embedded into tokens with pathology foundation models and then pooled into condition hidden vectors in **Stem**. Count values for each input gene is first scaled up by the gene count encoder and then combined with a trainable gene type embedding matrix. The backbone of **Stem** follows the design of DiT blocks and training scheme for **Stem** follows DDPM (see Sec 4 for more details).

pendent constants to simplify the diffusion design space, following the same practice as Ho et al. (2020), $\sigma_t^2 = \frac{1-\bar{\alpha}_{t-1}}{1-\bar{\alpha}_t}\beta_t$. We parameterize $\mu_\theta(X_t, V, t) = \frac{1}{\sqrt{\alpha_t}}(X_t - \frac{\beta_t}{\sqrt{1-\bar{\alpha}_t}}\epsilon_\theta(X_t, V, t))$, where $\epsilon_\theta(X_t, V, t)$ is a conditional noise prediction network that attempts to learn the noise $\epsilon_t$ contained in $X_t$ given the histology image patch $V$. With this setting, the diffusion model can be trained with a mean-squared error between the predicted noise $\epsilon_\theta(X_t, V, t)$ and the true Gaussian noise $\epsilon_t$:

$$\mathcal{L}_\epsilon(\theta) = \mathbb{E}_{t,x_t}\|\epsilon_\theta(X_t, V, t) - \epsilon_t\|_2^2$$

To improve training stability and avoid overfitting, we additionally augment the training dataset through image transformation. We consider simple transformations that will not distort the image quality and affect the embedding quality, such as rotations, flipping, and transposing. We ablate over the image augmentation technique in Sec. 5.3. Finally, to get a prediction for the gene expression values from $p_{\text{gene}}(X|V)$, we can build a statistical estimator using samples from this distribution. At inference time, when given a new histology image, we generate multiple samples using this histology image patch as condition and then take a sample mean over the generated gene expression vector to get a single prediction value.

**Multimodal Architecture Fusing Histology and Transcriptomics**   For a histology-conditional sampling of the corresponding gene expression profiles, our neural network model has to take both modality, histology images and gene expressions, into account. We use pre-trained pathology foundation models to derive suitable representations of histology image patches and design a special encoder to embed the gene expression vector into a sequence of latent embeddings. An overview of the architecture is presented in Fig.2.

Our architecture builds upon the Diffusion Transformer (DiT) architecture (Peebles & Xie, 2023) as well as the recent advances in computational pathology foundation models. DiT was originally designed for class conditional image generation and it uses a modulation mechanism to propagate the effect of timesteps of the diffusion model and inputted conditions across all layers. Similarly, we use the sinusoidal embedding of timestep $t$ and the latent embedding $c_{\text{hist}}$ of histology image patches as the input to the modulation module. We distill knowledge from state-of-the-art pathology foundation models such as UNI (Chen et al., 2024) and CONCH (Lu et al., 2024) through the computation of $c_{\text{hist}}$. We pass the histology image patch into foundation models to extract expressive token-level embeddings and perform attention pooling to get a single latent vector that aggregates the information of the image patch. We derive $c_{\text{hist}}$ by linearly projecting the latent vector to our desired model hidden dimension with an MLP.

To adapt DiT for our conditional gene expression generation task, we further remove the image-relevant-only modules in DiT, such as the image patchifier and unpatchifier. We treat the gene expression vector $X \in \mathbb{R}^C$ as a sequence of $C$ tokens, with each token taking value in $\mathbb{R}$. Since $X$ represents counts of different genes in the sample, we embed the count value and gene type respectively and aggregate through summation to build a unified embedding for each token. For the embedding of the $i$-th token, let $x_i$ denote the count value for the $i$-th gene in $X$, $1 \leq i \leq C$, and $h_i$

denote the embedding vector of the $i$-th token. We compute $h_i$ as,

$$h_i = h_i^{\text{count}} + h_i^{\text{type}}$$

where the count value embedding $h_i^{\text{count}} = \text{MLP}(x_i)$ is computed by passing $x_i$ through an MLP. The gene type embedding $h_i^{\text{type}}$ is a learnable embedding vector associated with the gene type of token $i$. The embedding of $X$ together with the timestep embedding and histology condition is then passed through a sequence of DiT transformer blocks. After the final DiT block, the sequence of gene tokens is decoded into an output noise prediction. For more details on neural network architecture, please refer to Appendix C.

## 5 Experimental Results

In this section, we present numerical results on Kidney Visium dataset, HER2ST dataset, and ablation studies on model hyperparameters and algorithm design choices. For additional results on other datasets, please refer to Appendix D.

**Evaluation metrics**   Our evaluation metrics include top $k$ mean Pearson Correlation Coefficient (PCC) (denoted as PCC-$k$, calculated in the log-transformed space), mean absolute error (MAE), mean square error (MSE), which have been widely used in evaluating the accuracy of gene expression prediction in the existing literature. MAE and MSE are calculated respectively using all genes in the selected gene set in log-transformed space.

As is pointed out in Xie et al. (2024), one pitfall in prediction tasks is to output the mean value with minimal variation or without meaningful variation that is faithful to the ground truth biological heterogeneity. Also, in our exploration, we discovered that high PCC does not necessarily guarantee meaningful and faithful predictions that align well with ground truth expression. In fact, we discovered that PCC in log-transformed space would be surprisingly high if the prediction is simply the mean expression across all genes in this spot. This encourages us to propose a new evaluation metric that can better reflect how well a prediction model captures the heterogeneity within the data. We consider the following relative variation distance (RVD), calculated through:

$$\text{RVD} = \frac{1}{C} \sum_{i=1}^{C} \frac{(\sigma_{\text{pred}}^{2,i} - \sigma_{\text{gt}}^{2,i})^2}{(\sigma_{\text{gt}}^{2,i})^2}$$

where $\sigma_{\text{pred}}^{2,i}$ is the variance of the $i$-th gene expression prediction across spots (predicted gene variation) and $\sigma_{\text{gt}}^{2,i}$ is the variance of the true $i$-th gene expression across spots (true gene variation). RVD represents a weighted average of the magnitude of deviation of the predicted gene variation from the true gene variation. RVD serves as a complementary metric to the current existing evaluation system that can better filter out false positive predictions created by solely focusing on PCC values. Additionally, we plot the gene variation curve against the ground truth, see Appendix A for more details.

### 5.1 Kidney Visium dataset

**Dataset and Preprocessing**   We applied **Stem** to a dataset that contains 23 kidney tissue sections from 22 individuals (Lake et al., 2023) covering three different health conditions (healthy reference (Ref), Chronic Kidney Disease (CKD), and Acute Kidney Injury (AKI)) and two different tissue types (cortex and medulla). Number of ST spots ranges from 315 to 4159 per slide. The gene profiling is performed using 10x Genomics Visium platform, which is a mainstream spatial transcriptomic sequencing platform that provides genomics profiling for a grid of spots with a diameter $\sim 55 \mu m$ along the tissue slide. We log-transformed the gene expression following Jaume et al. (2024).

**Experiment Setup**   An image patch of $224 \times 224$ pixels is cropped centered around each spot. Following a similar gene selection protocol in (He et al., 2020), we selected two gene sets, top 200 genes from the intersection of highly expressed (high mean) and highly variant (high variance) genes (denoted as HMHVG) and top 200 genes from all highly variable genes ordered in mean (denoted as HVG). Training, inference, and evaluation are performed in the log-transformed gene

count space to mitigate the impact of genes with extremely high expression counts. Evaluation of **Stem** is performed on the holdout slide, 20-0038 (AKI), which is randomly chosen from 23 slides. For experiment results on other holdout slides, please see details in the Sec. 5.3.

Table 1: Results on Kidney Visium dataset, compared with HisToGene (Pang et al., 2021), BLEEP (Xie et al., 2024), TRIPLEX (Chung et al., 2024). Higher values on PCC-10, PCC-50, PCC-200 are better. Lower values on MAE, MSE, RVD are better.

| | HMHVG | | | | | | HVG | | | | | |
|---|---|---|---|---|---|---|---|---|---|---|---|---|
| Model | PCC-10↑ | PCC-50↑ | PCC-200↑ | MAE↓ | MSE↓ | RVD↓ | PCC-10↑ | PCC-50↑ | PCC-200↑ | MAE↓ | MSE↓ | RVD↓ |
| HisToGene | 0.4294 | 0.3503 | 0.0905 | 0.9298 | 1.4105 | 0.9962 | 0.4237 | 0.3296 | 0.0774 | 0.9776 | 1.5609 | 0.9965 |
| BLEEP | 0.4998 | 0.4221 | 0.3143 | 0.9451 | 1.5261 | 0.2170 | 0.4902 | 0.3953 | 0.2474 | 0.9931 | 1.7658 | 0.3293 |
| TRIPLEX | 0.4654 | 0.4105 | 0.3165 | 0.8969 | **1.3015** | 0.5871 | 0.4621 | 0.3997 | 0.2726 | 0.9962 | **1.4500** | 0.6984 |
| **Stem** | **0.5893** | **0.5332** | **0.4257** | **0.8792** | 1.3513 | **0.0751** | **0.5366** | **0.4699** | **0.3047** | **0.9763** | 1.7529 | **0.1325** |

**Results**  As is shown in Table.1, **Stem** outperforms existing methods in almost all evaluation metrics for both HMHVG and HVG. Low RVD values indicate that **Stem** also preserves gene variations in inference compared to ground truth variations and successfully retains biological heterogeneity that resembles ground truth data. Since almost every slide in this dataset comes from a different patient with a distinct condition, the good performance indicates that our proposed approach is robust under batch effect and technical variations from the experimental side, and can generalize well to unseen histology images through predicting accurate gene expression values.

## 5.2 HER2ST DATASET

**Dataset and Preprocessing**  We also applied **Stem** to one breast cancer dataset, HER2ST (Andersson et al., 2021), which is sequenced by SpatialTranscriptomics[1] platform. This dataset includes 36 slices from 8 patients. From patient A-D, six tissue sections were collected with a distance of $32\mu$m in between. From patient E-H, three consecutive tissue sections were taken for each patient. Since intuitively it would be easier to infer gene expressions for consecutive slides if their neighbors are included in the training data, we make the task more challenging by holding out B1 (which does not have any neighboring consecutive slide), for test evaluation. Each slide contains normal tissue regions and some of the slides contain in situ cancer or invasive cancer. The spot size for this dataset is $100\mu$m in diameter and the total number of spots ranges from 176 to 712 per slide. We log-transformed the gene expression following Jaume et al. (2024).

**Experiment Setup**  An image patch of $224 \times 224$ is cropped around each spot. For the gene sets, we select top 300 HMHVG and 296 differentially expressed genes (DEGs), respectively, to perform training and evaluation. In HER2ST, the first slide in every patient is manually annotated by pathologists into 4 normal regions and 2 tumor regions. DEGs are selected following the standard preprocessing pipeline using *Scanpy* (Wolf et al., 2018) and the union of DEGs across all 6 regions from training slides are selected as features.

Table 2: Results on HER2ST dataset, compared with HisToGene(Pang et al., 2021), BLEEP(Xie et al., 2024), TRIPLEX (Chung et al., 2024). Higher values on PCC-10, PCC-50, PCC-300 are better. Lower values on MAE, MSE, RVD are better.

| | HMHVG | | | | | | DEG | | | | | |
|---|---|---|---|---|---|---|---|---|---|---|---|---|
| Model | PCC-10↑ | PCC-50↑ | PCC-300↑ | MAE↓ | MSE↓ | RVD↓ | PCC-10↑ | PCC-50↑ | PCC-300↑ | MAE↓ | MSE↓ | RVD↓ |
| HisToGene | 0.6812 | 0.6345 | 0.5250 | 0.9367 | 1.3468 | 10.3407 | 0.6816 | 0.6369 | 0.5112 | 0.8791 | 1.2627 | 9.7057 |
| BLEEP | 0.7727 | 0.7141 | 0.5652 | 0.8328 | 1.2428 | 0.6025 | 0.7711 | 0.7188 | 0.5518 | 0.7590 | 1.1297 | 0.6383 |
| TRIPLEX | 0.7907 | 0.7394 | 0.5766 | 0.9311 | 1.3456 | 0.6428 | 0.7919 | 0.7432 | 0.5709 | 0.8768 | 1.2887 | 0.6533 |
| **Stem** | **0.8298** | **0.7726** | **0.5984** | **0.7547** | **1.0742** | **0.0693** | **0.8365** | **0.7651** | **0.5748** | **0.6881** | **0.9631** | **0.0862** |

---

[1]In this paper, we use SpatialTranscriptomics platform to refer to one ST platform that was the first-ever appearance of ST introduced in 2016 (Ståhl et al., 2016)

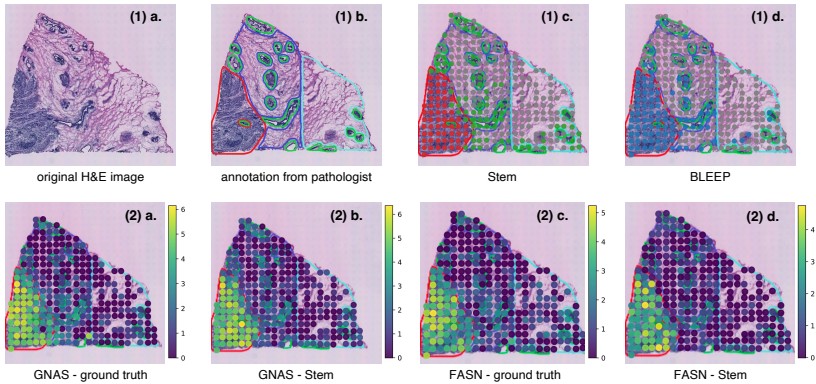

Figure 3: Visualization of unsupervised clustering results and cancer biomarker genes.

**Downstream Analysis of Unsupervised Tissue Structure Annotation**    As is shown in Table.2, **Stem** again surpassed all the existing methods across all evaluation metrics. Since we have valuable human annotations for this dataset, we could perform downstream analysis to further evaluate the model performance. Following the standard Leiden clustering pipeline in *Scanpy*, we obtained unsupervised clustering results based on the predicted 296 DEG expressions. Ideally, those genes are differentially expressed between different tissue structure regions, thus their gene expression pattern carries a certain level of information to distinguish different tissue regions. Fig.3 (1)c. and (1)d. shows Leiden clustering results based on **Stem**'s prediction and BLEEP's prediction. Other irrelevant cluster colors are suppressed and plotted as gray dots to better highlight the results. Leiden clustering algorithm results in two distinct clusters based on **Stem**'s prediction. Those two distinct clusters match the pathologist's annotation (Fig.3 (1)b.) for invasive cancer (red) and breast glands (green). In BLEEP's clustering results, invasive cancer and breast gland regions are clustered into the same group based on BLEEP's gene expression prediction. This illustrates that **Stem**'s inference accurately aligns with the ground truth biology. We also highlighted **Stem**'s prediction for two well-known cancer markers, GNAS and FASN (He et al., 2020), and their ground truth expression level in Fig.3. For visualization of other cell-type-specific marker genes identified in Andersson et al. (2021), see Appendix E. It's worth mentioning that both the gene expression pattern as well as the scale of expression intensity (shown in the colorbar) match well between **Stem**'s prediction and ground truth, which further demonstrates the power of **Stem**.

## 5.3    ABLATION STUDY

In this section, we perform an ablation study on the model hyperparameters and algorithm design choices. We examine the following three factors in order: choices of pathology foundation model, image augmentation ratio, and test slide health condition. In the following, we perform all the experiments on the Kidney Visium dataset and use the HMHVG gene set. We set the default setting to be: CONCH + UNI for the pathology foundation model, $1:4$ for image augmentation ratio, and 20-0038 (AKI) for the hold-out test slide. Unless further notice is given, we will keep the setting the same as the default and only vary the ablated parameter for each ablation study. The best values are marked in **bold**. We also perform additional ablation experiments on the scalability of **Stem** to large gene sets, effects of generated samples and sample statistics, influence of pathology foundation model size, and the representation power of histology image patch encoder. For more details on the extra ablation experiments, please refer to Appendix B.

**Choice of Foundation Models**    In this ablation experiment, we specifically choose not to augment the training dataset with image augmentation techniques. We seek to compare the effects of foundation models on algorithm performance by removing other potential influencing factors. We consider the possible combination of CONCH (Lu et al., 2024) and UNI (Chen et al., 2024), which produces embedding vectors of dimension 512 and 1024 for histology image patch of size $224 \times 224$ respectively. We evaluate the following three sets of choices: 1) CONCH only 2) UNI only 3) CONCH + UNI, and the evaluation result is in Table 3. Here, CONCH + UNI stands for using combined features extracted by both UNI and CONCH through simple concatenation. The results suggest that it works best to input the model combined histology image information of both foundation models. This is reasonable and accords with our intuition since UNI and CONCH are trained with distinct

self-supervised techniques, and thus are capable of extracting different types of features from the same histology images. The combined embedding provides the neural network with richer information than using UNI or CONCH alone and leads to better numerical performances.

Table 3: Results of ablation study on choice of pathology foundation model. Higher values on PCC-10, PCC-50, PCC-200 are better. Lower values on MAE, MSE, RVD are better.

| Foundation Model | HMHVG | | | | | |
| --- | --- | --- | --- | --- | --- | --- |
| | PCC-10↑ | PCC-50↑ | PCC-200↑ | MAE↓ | MSE↓ | RVD↓ |
| CONCH | 0.3662 | 0.3117 | 0.2254 | 1.0768 | 1.9952 | **0.0625** |
| UNI | 0.4690 | 0.4340 | 0.3288 | 0.9161 | 1.4588 | 0.1561 |
| **CONCH + UNI** | **0.4817** | **0.4359** | **0.3289** | **0.9135** | **1.4336** | 0.1016 |

**Image Augmentation**    In this ablation experiment, we found that additional augmentation of the training dataset by pairing each gene expression with a transformed version of the original histology image could significantly boost the algorithm's performance. We randomly transform the histology image with the following 7 transformations: horizontal flip, vertical flip, 90-degree rotation, 180-degree rotation, 270-degree rotation, transpose, and transverse. We select these transformations since they do not distort the histology images and cause information loss.

We varied the size ratio between the original dataset and the synthetically augmented dataset in increasing order from 2:1 to 1:4. The evaluation result is in Table 4. We see that the algorithm benefits from having more synthetically augmented training data, although the gain quickly saturated as we increase the augmentation ratio. Judging from the metrics, 1:4 seems to be the best setting, while 1:2 also shows a compelling performance on the gene variation distance.

Table 4: Results of ablation study on the choice of image augmentation ratio. Higher values on PCC-10, PCC-50, PCC-200 are better. Lower values on MAE, MSE, RVD are better.

| Ratio | HMHVG | | | | | |
| --- | --- | --- | --- | --- | --- | --- |
| | PCC-10↑ | PCC-50↑ | PCC-200↑ | MAE↓ | MSE↓ | RVD↓ |
| 2:1 | 0.4843 | 0.4338 | 0.3298 | 0.9375 | 1.5147 | 0.1058 |
| 1:1 | 0.5124 | 0.4622 | 0.3532 | 0.9119 | 1.4350 | 0.1391 |
| 1:2 | 0.5373 | 0.4872 | 0.3832 | 0.9098 | 1.4413 | **0.0813** |
| **1:4** | **0.5485** | **0.4947** | **0.3859** | **0.8962** | **1.3982** | 0.1316 |

**Test Slide Health Condition**    Finally, we ablate over the influence of choosing holdout slides with different health conditions. The evaluation result is in Table 5. We note that **Stem** performs similarly when the two disease condition slides, AKI and CKD, are used for inference.

Table 5: Results of ablation study on the choice of test slide under different health conditions. AKI: Acute Kidney Injury. CKD: Chronic Kidney Disease. Higher values on PCC-10, PCC-50, PCC-200 are better. Lower values on MAE, MSE, RVD are better.

| Holdout Slide | HMHVG | | | | | |
| --- | --- | --- | --- | --- | --- | --- |
| | PCC-10↑ | PCC-50↑ | PCC-200↑ | MAE↓ | MSE↓ | RVD↓ |
| 20-0038 (AKI) | 0.5893 | 0.5332 | 0.4257 | 0.8792 | 1.3513 | 0.0751 |
| 20-0071 (AKI) | 0.5685 | 0.5263 | 0.4239 | 0.9316 | 1.4551 | 0.0807 |
| 21-0057 (CKD) | 0.7026 | 0.5954 | 0.4502 | 0.9758 | 1.5422 | 0.1140 |

# 6    CONCLUSION AND FUTURE WORK

In this work, we propose **Stem**, a novel generative modeling algorithm for spatially resolved gene expression prediction based on H&E stained histology images using conditional diffusion models. **Stem** generates highly accurate and biologically faithful predictions for unseen histology images at test time and achieves SOTA performance on multiple evaluation metrics across different datasets. For future work, it would be exciting to explore more conditioning mechanism, neural network architecture, and their influence on task performance. How to better use embedding generated by pathology foundation models on diffusion generative modeling is also an intriguing question for future explorations.

**Acknowledgement**    YZ and MT are grateful for partial support by NSF DMS-1847802.

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

# A  GENE VARIATION CURVES FOR KIDNEY VISIUM AND HER2ST DATASET

In this section, we present the gene variation comparison curves between predictions and ground truth values for each gene set. In each of these plots, the x-axis is the index for every predicted gene ordered by either ground truth variance normalized over the sum of total ground truth variance (top row) or the absolute ground truth variance without normalization (bottom row). The blue curve shows the ground truth value of gene variance while orange dots are predicted gene variations ordered from low to high in their ground truth variance. A smaller distance from the orange dots to the blue line indicates a better recovery of gene variations. We compare **Stem** with TRIPLEX (Chung et al., 2024), BLEEP (Xie et al., 2024), and HisToGene (Pang et al., 2021). Compared with existing approaches, **Stem** has prediction variation closer to the ground truth variation curve with a smaller degree of dispersion.

## A.1  KIDNEY VISIUM DATASET

Results for the HMHVG gene set are in Fig.4 and results for the HVG gene set are in Fig.5.

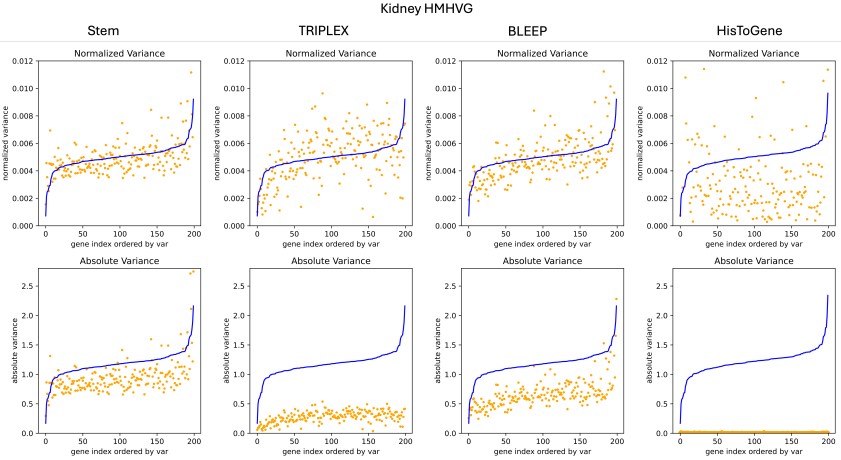

Figure 4: Gene variation comparison between prediction and ground truth for HMHVGs in the Kidney Visium dataset. A closer match to the blue curve is better.

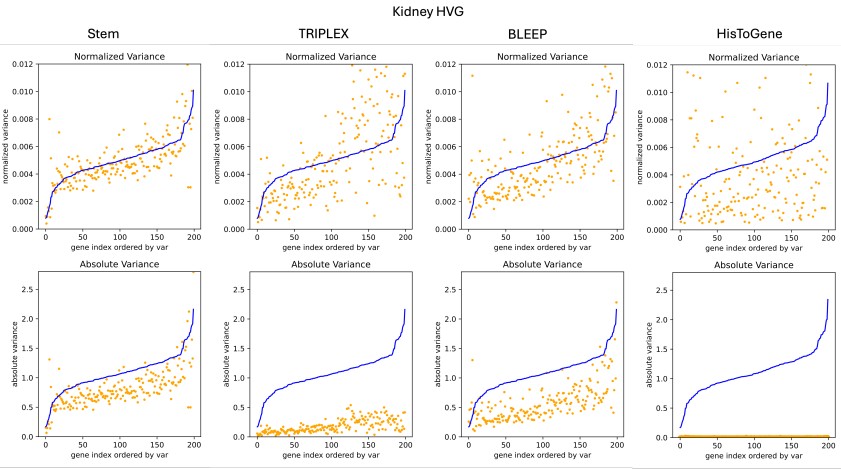

Figure 5: Gene variation comparison between prediction and ground truth for HVGs in the Kidney Visium dataset. A closer match to the blue curve is better.

## A.2 HER2ST

Results for the HMHVG gene set are in Fig.6 and results for the DEG gene set are in Fig.7.

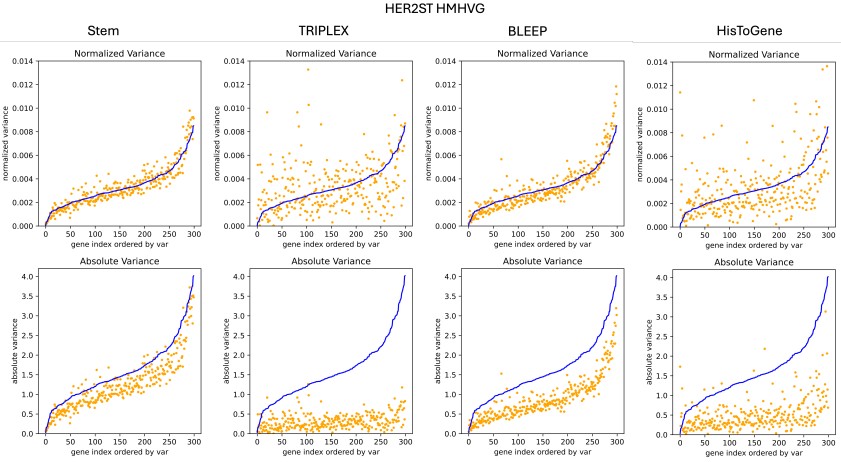

Figure 6: Gene variation comparison between prediction and ground truth for HMHVGs in the HER2ST dataset. A closer match to the blue curve is better.

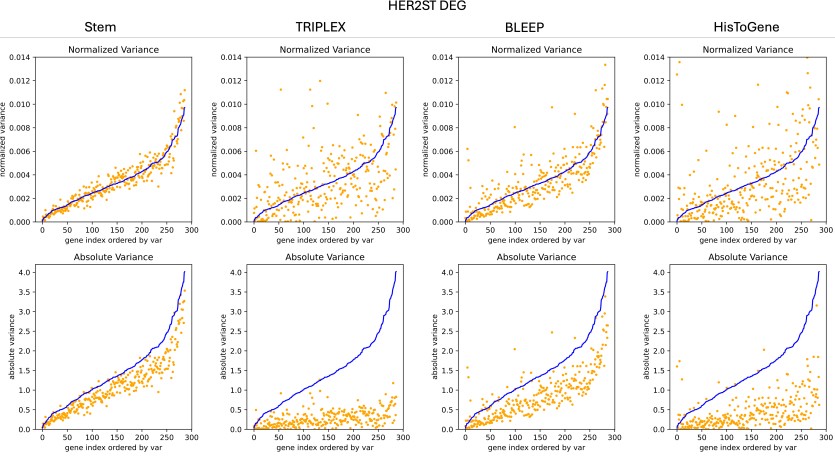

Figure 7: Gene variation comparison between prediction and ground truth for DEGs in the HER2ST dataset. A closer match to the blue curve is better.

## B    ADDITIONAL ABLATION STUDY

In this section, we present additional results for ablation experiments to further testify the algorithmic robustness of **Stem**. We probe into the following four questions in order: scalability of **Stem** to large gene sets, effects of generated samples and sample statistics, influence of pathology foundation model size, and the representation power of histology image patch encoder. The ablation experiments for the first two questions are performed on the HER2ST dataset while the experiments for the latter two questions are performed on the Kidney Visium dataset. Across all experiments, we choose the following default algorithm design choice for **Stem**: CONCH + UNI for the pathology foundation model, and 1:4 for the image augmentation ratio. The holdout test slide is B1 for HER2ST experiments and 20-0038 (AKI) for Kidney Visium experiments.

**Large gene sets**    In this ablation experiment, we evaluate the scalability of **Stem** by testing on a large gene set. We select 1000 Differentially Expressed Genes (DEGs) from the HER2ST dataset.

The evaluation result is shown in Table 6. We notice that **Stem** consistently outperforms other approaches in almost all metrics, suggesting strong scalability of our proposed approach in terms of predicted gene panel size. Notably, **Stem** still achieves a low RVD value in this case, which demonstrates that it is capable of learning the complicated spatial heterogeneity even when simultaneously predicting a large number of genes.

Table 6: Results of ablation study on scalability of **Stem** to large gene sets, compared with HisTo-Gene (Pang et al., 2021), BLEEP (Xie et al., 2024), TRIPLEX (Chung et al., 2024). Higher values on PCC-10, PCC-50, PCC-300, PCC-1000 are better. Lower values on MAE, MSE, RVD are better.

| | DEG | | | | | | |
|---|---|---|---|---|---|---|---|
| Model | PCC-10↑ | PCC-50↑ | PCC-300↑ | PCC-1000↑ | MAE↓ | MSE↓ | RVD↓ |
| HisToGene | 0.6136 | 0.5784 | 0.5061 | 0.2301 | 0.9811 | 1.3283 | 0.9958 |
| BLEEP | 0.7485 | 0.7016 | 0.6307 | 0.5058 | 0.6170 | 0.8675 | 0.5220 |
| TRIPLEX | 0.7924 | 0.7558 | 0.6843 | **0.5575** | 0.6781 | 0.8143 | 0.6632 |
| **Stem** | **0.8279** | **0.7797** | **0.6980** | 0.5423 | **0.5649** | **0.7471** | **0.1208** |

**Generated Samples and Sample Statistics**     In this ablation experiment, we aim to investigate the effects of different samples generated using **Stem**, as well as the influence of the statistics function used to summarize these samples for predictions. In this experiment, **Stem** is trained on 1000 DEGs from the HER2ST dataset. We generate 100 samples for each test histology image patch and compute several different statistics given those 100 generated samples. We visualize all 100 generated samples, three different sample statistics, and the ground truth value for 6 randomly selected gene pairs in Fig.8. Apart from the simple sample mean, we also compute the sample median (the value of 50% quantile of the generated samples) and the sample mode (the value with the highest probability among the generated samples), for a more comprehensive comparison. We also include the evaluation metrics with these sample statistics as predictions in Table 7.

From Fig.8, we can see that the generated samples cluster around the ground truth values. All the chosen statistics functions manage to summarize well the generated samples and produce predictions with a reasonably small distance to the ground truth value. Results in Table 7 suggest that, overall, the sample mean achieves the best numerical performance, and thus we choose the sample mean to be the predicted gene expression value. However, we do notice that there are situations where other statistics perform better than the sample mean. For example, in Fig.8(b), the sample median exactly overlaps with the ground truth value, while the sample mean has the largest distance to the ground truth. Therefore, we believe it's possible to design better sample statistics to further boost the performance of **Stem**, which we plan to investigate in future works.

Table 7: Results of ablation study on sample statistics. Higher values on PCC-10, PCC-50, PCC-300, PCC-1000 are better. Lower values on MAE, MSE, RVD are better.

| | DEG | | | | | | |
|---|---|---|---|---|---|---|---|
| Statistics | PCC-10↑ | PCC-50↑ | PCC-300↑ | PCC-1000↑ | MAE↓ | MSE↓ | RVD↓ |
| Median | 0.8222 | 0.7738 | 0.6871 | 0.5263 | **0.5616** | 0.7909 | 0.0950 |
| Mode | 0.8204 | 0.7715 | 0.6839 | 0.5222 | 0.5620 | 0.8030 | **0.0928** |
| Mean | **0.8279** | **0.7797** | **0.6980** | **0.5423** | 0.5649 | **0.7471** | 0.1208 |

**Large Pathology Foundation Models**     In this ablation experiment, we test the influence of pathology foundation model sizes on the performance of **Stem**. Apart from CONCH (0.1 Billion parameters) (Lu et al., 2024) and UNI (0.3 Billion parameters) (Chen et al., 2024), we additionally select two larger pathology foundation models, Virchow-2 (0.6 Billion parameters) (Zimmermann et al., 2024) and H-Optimus-0 (1.1 Billion parameters) (Saillard et al., 2024), and benchmark the performance of **Stem** on the Kidney Visium dataset with those four foundation models being the histology image patch encoder. The evaluation result is presented in Table 8. Among the four foundation models mentioned above, UNI, Virchow-2, and H-Optimus-0 are vision-only models and are trained using DINOv2 (Oquab et al., 2023), while CONCH differs from them by being a Vision-Language Model (VLM) and is trained using contrastive image captioning loss following CoCa (Yu et al., 2022).

Judging from the numerical results, we observe that larger pathology foundation models do not necessarily imply a better performance for **Stem**. While our proposed algorithm is robust to the choice of foundation models as using Virchow-2 and H-Optimus-0 also produces satisfactory performance, these results are subpar compared with only using UNI. This suggests that model size might not be

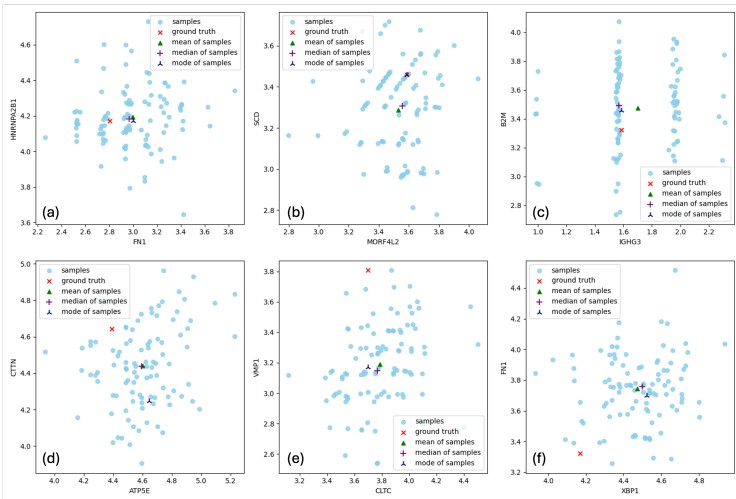

Figure 8: Visualization of generated samples and computed sample statistics. Two axes are two randomly selected genes from the DEG gene set. Skyblue dots: samples generated by **Stem**. Red marker: ground truth. Green marker: sample mean. Purple marker: sample median. Navy marker: sample mode.

the deciding factor when choosing foundation models as good patch encoders. We also note that using UNI/Virchow-2/H-Optimus-0 alone leads to worse performance than using CONCH + UNI in all metrics. This observation re-emphasizes the importance of patch embedding diversity, which can be achieved through using CONCH in addition to UNI. It also further suggests the possibility of achieving even better performance with **Stem** by using different combinations of foundation models (such as H-Optimus-0 + Virchow + CONCH), which we plan to investigate in future works.

Table 8: Results of ablation study on the influence of pathology foundation model sizes. Higher values on PCC-10, PCC-50, PCC-200 are better. Lower values on MAE, MSE, RVD are better.

| Model (Stem + ) | PCC-10↑ | PCC-50↑ | PCC-200↑ | MAE↓ | MSE↓ | RVD↓ |
|---|---|---|---|---|---|---|
| | | | HMHVG | | | |
| UNI | 0.5593 | 0.5095 | 0.4093 | 0.8834 | 1.3690 | 0.1081 |
| Virchow-2 | 0.5404 | 0.4860 | 0.3683 | 0.9230 | 1.4703 | 0.1027 |
| H-Optimus-0 | 0.5435 | 0.4862 | 0.3735 | 0.9021 | 1.4092 | 0.1298 |
| CONCH + UNI | **0.5893** | **0.5332** | **0.4257** | **0.8792** | **1.3513** | **0.0751** |

**Power of Histology Image Patch Encoder**     Finally, we investigate the power and contribution of histology image patch encoders to the overall performance of **Stem**. We aim to demonstrate that while our proposed framework **Stem** potentially benefits from a strong image patch encoder, its success can't be solely attributed to good histology embeddings. Other algorithm components, such as diffusion models, are also essential contributing factors to the SOTA performances of **Stem**. To demonstrate this, we benchmark several common image patch encoders in the literature. Apart from pathology foundation models such as UNI and CONCH, we also experimented with ResNet18 trained on pathology images using contrastive learning (Ciga et al., 2022), which is also the image patch encoder used in TRIPLEX.

We build a simple but effective pipeline to generate gene expression predictions based solely on these pretrained histology image patch encoders, following a similar design as BLEEP. At the inference time, we encode the test image patch and retrieve its nearest neighbors from the training dataset, and then the averaged gene expression of these selected neighbors is used as the gene expression prediction for this test patch. Following this protocol, we evaluate the performance of these encoders on the HMHVG gene set of the Kidney Visium dataset and compare them with the results achieved by **Stem** in Table 9. Interestingly, ResNet18 and UNI perform the best under this setting and consistently generate better predictions than the setting of CONCH + UNI. However, the performance of **Stem** using combined UNI and CONCH still surpasses other approaches by a great margin. This indicates that **Stem** achieves a nontrivial improvement on the top of these encoders, thanks to the overall superiority of the algorithmic framework based on generative modeling and

conditional diffusion models. The improvement is most significant in terms of RVD, which implies that **Stem** does a particularly good job of recovering spatial biological heterogeneity.

Table 9: Results of ablation study on the power of histology image patch encoders. The performance of **Stem** is placed in the last row for a clear comparison between the power of histology image patch encoder and our proposed generative pipeline via conditional diffusion models. Higher values on PCC-10, PCC-50, PCC-200 are better. Lower values on MAE, MSE, RVD are better.

| | HMHVG | | | | | |
|---|---|---|---|---|---|---|
| Model | PCC-10↑ | PCC-50↑ | PCC-200↑ | MAE↓ | MSE↓ | RVD↓ |
| ResNet18 | **0.4795** | **0.3999** | 0.2297 | 1.0124 | 1.7687 | 0.4064 |
| CONCH | 0.3824 | 0.3250 | 0.2442 | 0.9805 | 1.5618 | **0.2687** |
| UNI | 0.4328 | 0.3779 | **0.2909** | **0.9012** | **1.3785** | 0.3599 |
| CONCH + UNI | 0.3954 | 0.3417 | 0.2547 | 0.9301 | 1.4506 | 0.3269 |
| **Stem (CONCH + UNI)** | 0.5893 | 0.5332 | 0.4257 | 0.8792 | 1.3513 | 0.0751 |

## C  NEURAL NETWORK ARCHITECTURE AND TRAINING SETUPS

**Neural Network Architecture**    Our neural network parameterizes the score function in diffusion models and is built on the top of DiT (Peebles & Xie, 2023). Additionally, we introduce a trainable gene encoder to embed the gene expression vectors into a sequence of latent embedding vectors as inputs to the diffusion transformer blocks. Specifically, the $i$-th gene is encoded into vector $h_i \in \mathbb{R}^D$ of hidden dimension $D$, using the following expression,

$$h_i = h_i^{\text{count}} + h_i^{\text{type}}$$

where $h_i^{\text{count}}$ is computed through passing the $i$-th scalar count into a 2-layer MLP with input dimension 1 and output dimension $D$, and $h_i^{\text{type}}$ is a learnable embedding vector of dimension $D$ associated with the $i$-th gene modeled using a look-up table. Apart from the gene encoding, we also embed time using the standard practice of 256-dim sinusoid embedding as in Dhariwal & Nichol (2021), followed by a 2-layer MLP with hidden dimension $D$ as in Peebles & Xie (2023).

For the histology image patch embedding, we follow the recommended practice given by the producer of each pathology foundation model to generate one single embedding vector per model for a given image patch. This is typically achieved by using the embedding vector of a special token or performing average or attention pooling to the sequence of image token embedding vectors. When using multiple foundation models, the extracted patch embedding is post-processed through simple concatenation and then fed into a 2-layer learnable MLP to obtain a true image conditioning vector of dimension $D$. The image conditioning vector enters the transformer blocks together with the sinusoid time embedding, through the modulation module realized by the adaptive LayerNormalization Layers (adaLN). Moreover, these layers are zero-initialized for more training benefits (Peebles & Xie, 2023).

After the final DiT block, the DiT-block output is fed into a simple output module as the decoder. The output module consists of one adaLN layer and a linear layer, and the linear layer has output dimension 1 as the diffusion model desires. This is also the same practice considered in Peebles & Xie (2023).

For all of our numerical experiments, we use 12 DiT blocks (with adaLN-Zero design), with 6-head attention and hidden dimension $D = 384$. For all the MLP mentioned above, we use SiLU as activation functions.

**Training Hyperparameters**    We train the neural network with AdamW optimizer, with a constant learning rate of $1 \times 10^{-4}$. We train all the models for 250k iterations with a batch size of 256, where the model typically converges after 150k iterations. We also adopt an Exponential Moving Average module (EMA) with a decay rate of 0.9999. During the inference phase, we produce predictions using 20 generated samples for each image patch.

## D  ADDITIONAL EXPERIMENTS AND DATASETS

To further demonstrate the robustness of **Stem** to datasets, we experiment on two additional Visium datasets with different organs and species from the Kidney Visium and HER2ST datasets. We consider a cancer human prostate dataset and a healthy mouse brain dataset, both gene profiling are

performed using Visium. We run **Stem** with CONCH + UNI as histology image patch encoder and a 1:4 image augmentation ratio, the same setting as the main results for Kidney Visium and HER2ST. Similarly, we compute evaluation metrics such as the top-$k$ Pearson correlation, MSE, MAE, and RVD.

## D.1 HUMAN PROSTATE CANCER (PRAD) VISIUM DATASET

**Dataset and Preprocessing**  We evaluate Stem on the prostate cancer Visium dataset (PRAD) which contained 23 Visium samples from 2 patients (Erickson et al., 2022). Both patients were diagnosed with prostatic acinar adenocarcinoma with a (4+3) Gleason score (ISUP group 4). The number of spots in one tissue slide ranges from 1418 to 4079 and the spot size is $55\mu$m. An image patch of $224 \times 224$ is cropped around each spot. For the selected gene set, we choose top 200 HMHVGs from the union of highly variable genes in each slide. We randomly choose and hold out the slide with ID MEND145 in the HEST-1k database (patient_2_V1_2 in the original dataset) as the test slide. We log-transformed the gene expression following Jaume et al. (2024)

**Results**  The numerical result is presented in Table 10. **Stem** achieves the best performance on almost all metrics compared with other regression-based approaches, with an especially large margin in RVD. We also present the gene variation curves of each method on this dataset in Fig.9. Note that while TRIPLEX produces competitive numbers in terms of Pearson correlation, its gene variation curves are almost flat. This implies that TRIPLEX does not output spatially diversified predictions for different histology patches, which is not ideal for the task of predicting gene expressions from H&E stained images. A similar situation happens to HisToGene as well.

Table 10: Results on the cancer human prostate Visium dataset, compared with HisToGene (Pang et al., 2021), BLEEP (Xie et al., 2024), TRIPLEX (Chung et al., 2024). Higher values on PCC-10, PCC-50, PCC-200 are better. Lower values on MAE, MSE, RVD are better.

| Model | HMHVG | | | | | |
|---|---|---|---|---|---|---|
| | PCC-10↑ | PCC-50↑ | PCC-200↑ | MAE↓ | MSE↓ | RVD↓ |
| HisToGene | 0.4035 | 0.3554 | 0.2235 | 0.9538 | **1.4619** | 0.8855 |
| BLEEP | 0.5798 | 0.5102 | 0.3158 | 1.0909 | 2.4754 | 0.4202 |
| TRIPLEX | **0.6173** | 0.4953 | 0.3601 | 0.9747 | 1.4819 | 0.7954 |
| **Stem** | 0.6103 | **0.5315** | **0.3832** | **0.8585** | 1.4873 | **0.1975** |

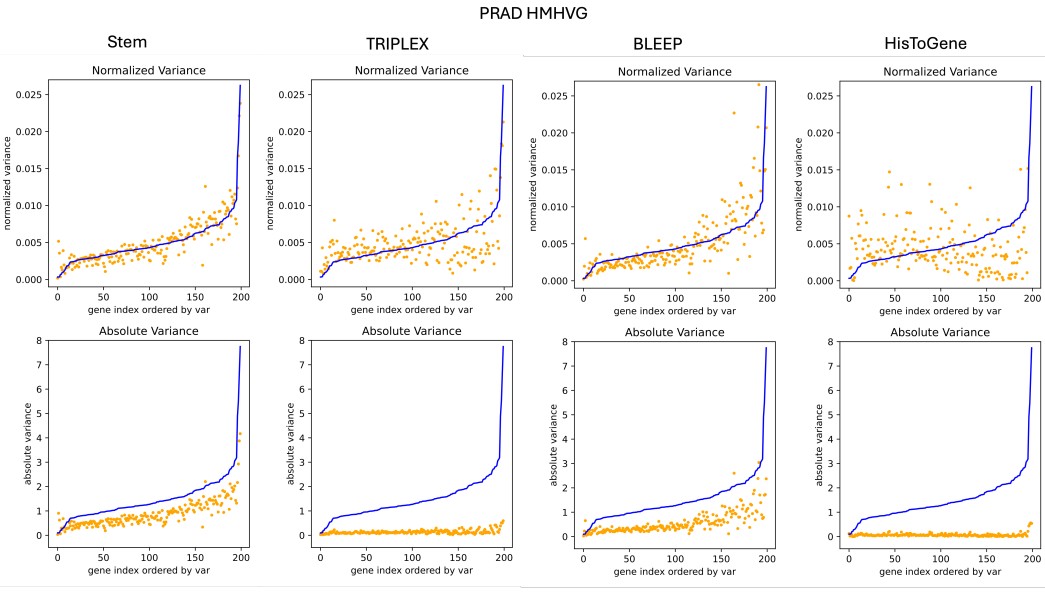

Figure 9: Gene variation comparison between prediction and ground truth for HMHVGs in the human prostate dataset. A closer match to the blue curve is better.

## D.2 HEALTHY MOUSE BRAIN DATASET

**Dataset and Preprocessing** We also evaluate **Stem** on one healthy mouse brain Visium dataset, which contains 14 Visium samples from 4 healthy adult male mice (Vicari et al., 2024). The number of spots in one tissue slide ranges from 2675 to 3617 and the spot size is $55\mu$m. An image patch of $224 \times 224$ is cropped around each spot. For the selected gene set, we choose top 200 HMHVGs from the union of highly variable genes in each slide. We randomly chose and held out the slide with ID NCBI667 in HEST-1k database as the test slide (ID in the original dataset: V11L12-109_A1). We log-transformed the gene expression following Jaume et al. (2024).

**Results** The numerical result is presented in Table 11. Again, **Stem** excels and shows a more appealing numerical performance than other methods in most of the metrics, despite the difficulty of this dataset. We include the gene variation curves of each method on this dataset in Fig.10. As is clear from the figure, **Stem** produces a good match with the truth gene variation curve in terms of both normalized and absolute variance. On this dataset, we fail to evaluate TRIPLEX potentially due to a limited GPU memory budget and thus the evaluation metrics are not presented.

Table 11: Results on the healthy mouse brain Visium dataset, compared with HisToGene (Pang et al., 2021), BLEEP (Xie et al., 2024), TRIPLEX (Chung et al., 2024). Higher values on PCC-10, PCC-50, PCC-200 are better. Lower values on MAE, MSE, RVD are better.

| | HMHVG | | | | | |
|---|---|---|---|---|---|---|
| Model | PCC-10↑ | PCC-50↑ | PCC-200↑ | MAE↓ | MSE↓ | RVD↓ |
| HisToGene | 0.3032 | 0.1665 | -0.0008 | **0.8983** | **1.2646** | 0.9236 |
| BLEEP | 0.3419 | 0.2799 | 0.1555 | 0.9872 | 1.5905 | 0.2385 |
| TRIPLEX[2] | \ | \ | \ | \ | \ | \ |
| **Stem** | **0.4908** | **0.4106** | **0.2791** | 0.9307 | 1.4752 | **0.0693** |

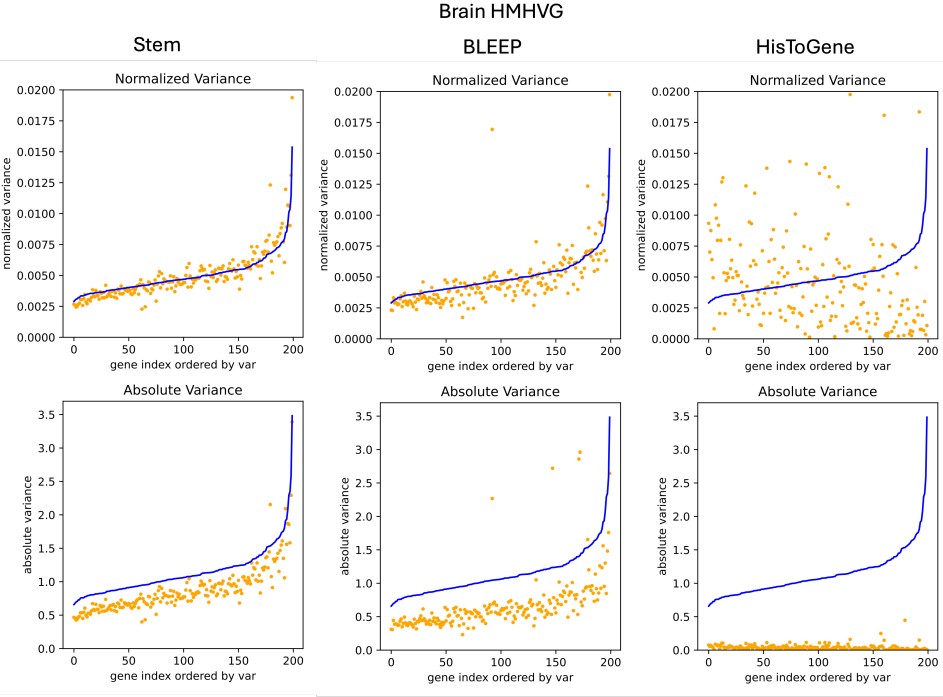

Figure 10: Gene variation comparison between prediction and ground truth for HMHVGs in healthy mouse brain dataset. A closer match to the blue curve is better.

---

[2]We fail to run TRIPLEX due to computational resource limits. TRIPLEX suffers from increasing GPU memory usage as the epoch number increases on the healthy mouse brain dataset.

## E  VISUALIZATION OF MARKER GENES IN HER2ST

In this section, we visualize predictions of more marker genes in the HER2ST dataset (using slide B1 and G2, which contain tissue annotations from the pathologists) and compare **Stem** against Hist2Gene, BLEEP, and TRIPLEX, the ground truth gene expressions and human annotations. We plot the predicted gene expressions using heatmaps overlaid on the whole H&E image for the following marker genes: CCL19, TRAC, IGHA1, GPX3, RAB11FIP1, COL4A1, which are identified in the original work of HER2ST. The authors of HER2ST identified CCL19 and TRAC as marker genes for immune cells (APC, B-cell, T-cell), IGHA1 as a marker gene for B/plasma cells, GPX3 as a marker gene for adipose tissue, RAB11FIP1 as a marker gene for immune rich in situ cancer, and COL4A1 as a marker gene for a mixture of cancer and connective tissue. We present the results for CCL19 in Fig.11, TRAC in Fig.12, IGHA1 in Fig.13, GPX3 in Fig.14, RAB11FIP1 in Fig.15 and COL4A1 in Fig.16.

As is evident from the figures, **Stem** manages to generate gene expression predictions with a pattern highly resembling the ground truth, while other algorithms fail to do so. One major issue that existing approaches suffer from is that they frequently overestimate the expression values for genes that are supposed to be sparsely expressed and underestimate the genes that might have high expression values in certain spots. For example, see the predictions of TRIPLEX for GPX3 in Fig.14, HisToGene for CCL19 in Fig.11, and BLEEP for COL4A1 in Fig.16. We also notice that **Stem** consistently produces predictions for marker genes with a scale close to the ground true scale, which can be seen by comparing the color bar of **Stem** predictions with that of ground truth values. However, other methods often generate predictions with a significant difference from the ground truth data, which indicates that the predictions are of low fidelity. An accurate prediction of cell-type-specific marker genes enables meaningful downstream tasks such as cell type identification, automated tissue region annotations, etc. These observations prove again that **Stem** excels in generating highly accurate gene expression profiles from histology images and capturing the spatial heterogeneity of ST data, paving the way for downstream analysis.

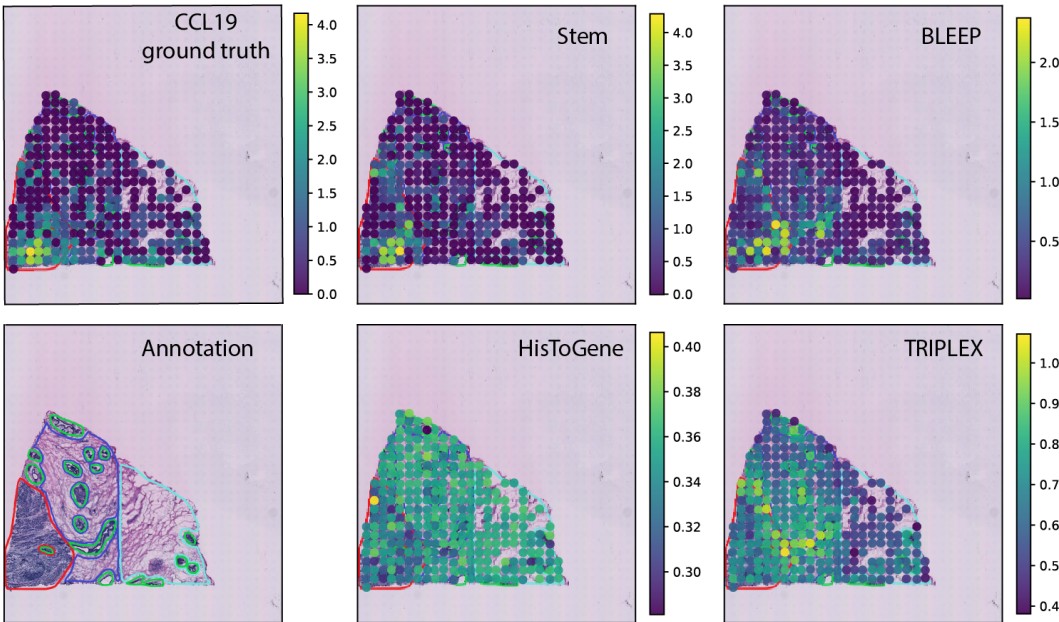

Figure 11: Visualization of marker gene CCL19 predictions

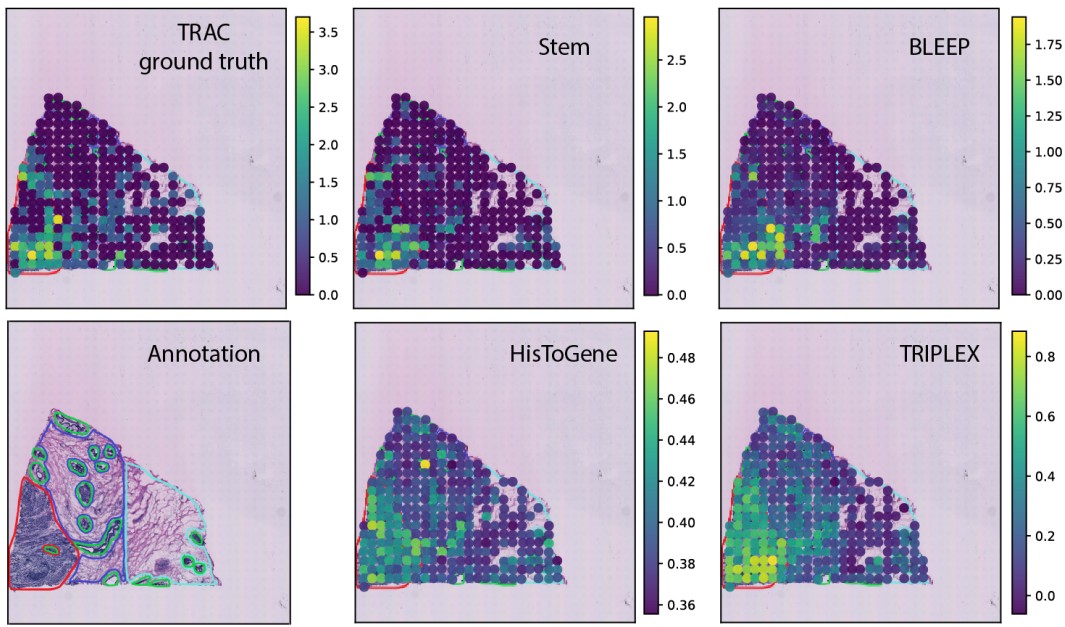

Figure 12: Visualization of marker gene TRAC predictions

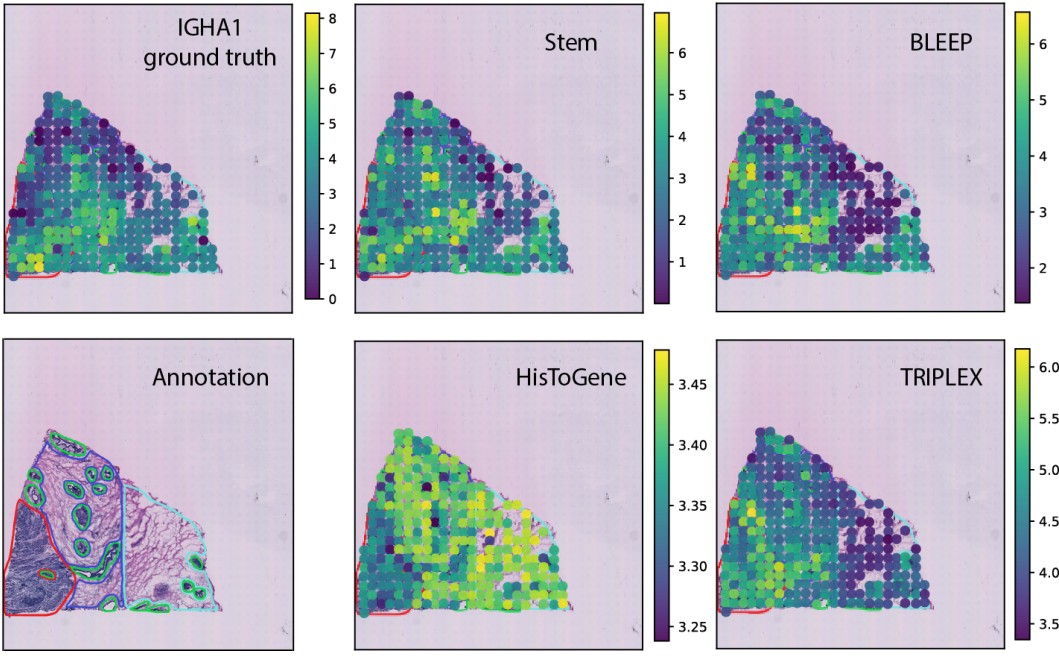

Figure 13: Visualization of marker gene IGHA1 predictions

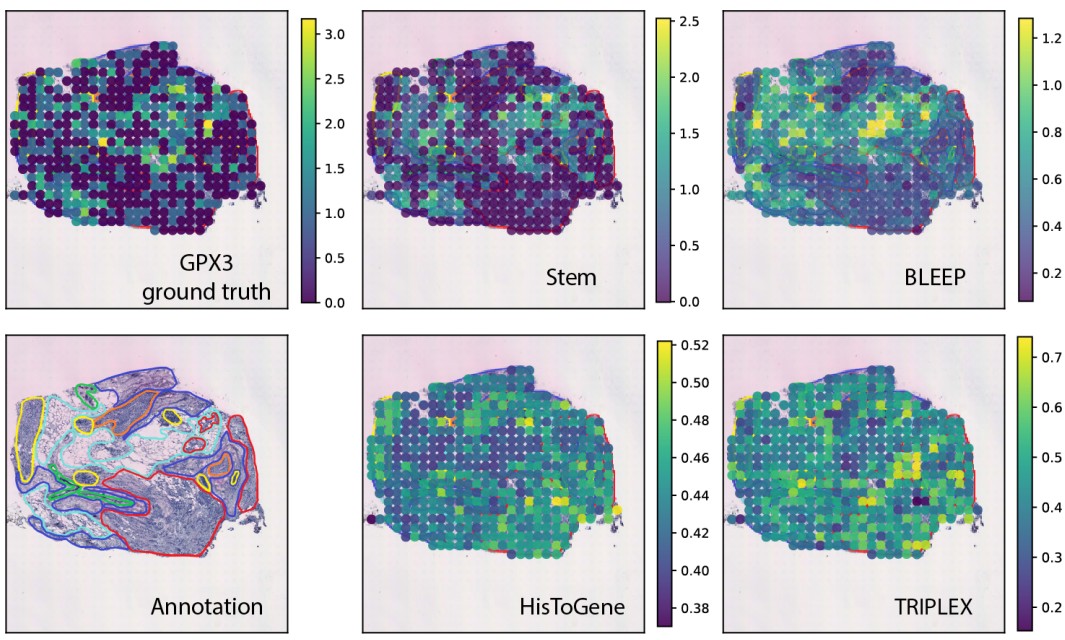

Figure 14: Visualization of marker gene GPX3 predictions

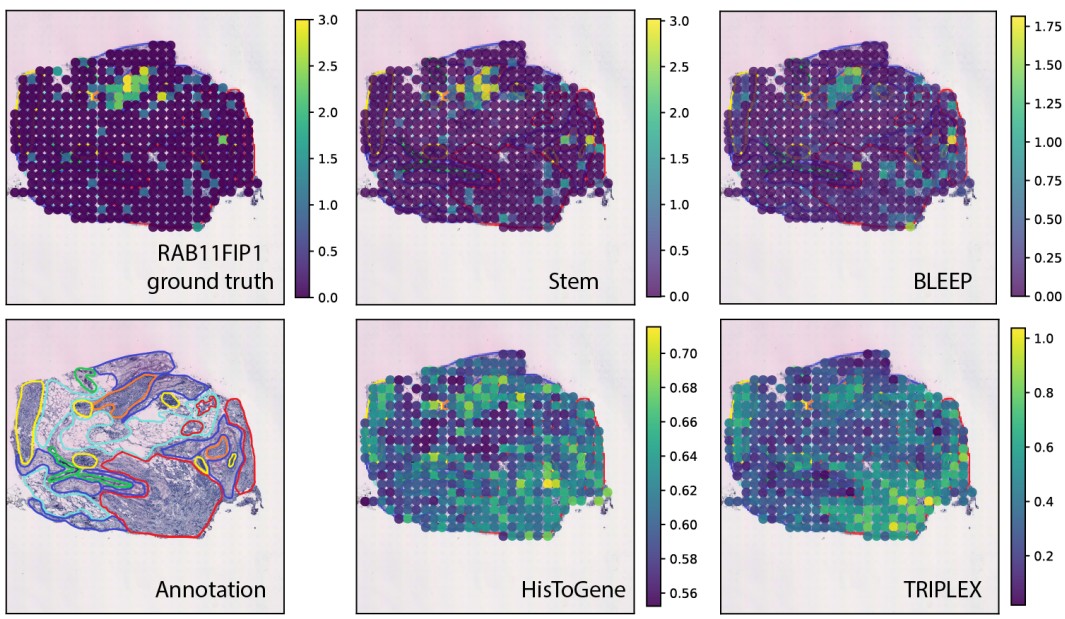

Figure 15: Visualization of marker gene RAB11FIP1 predictions

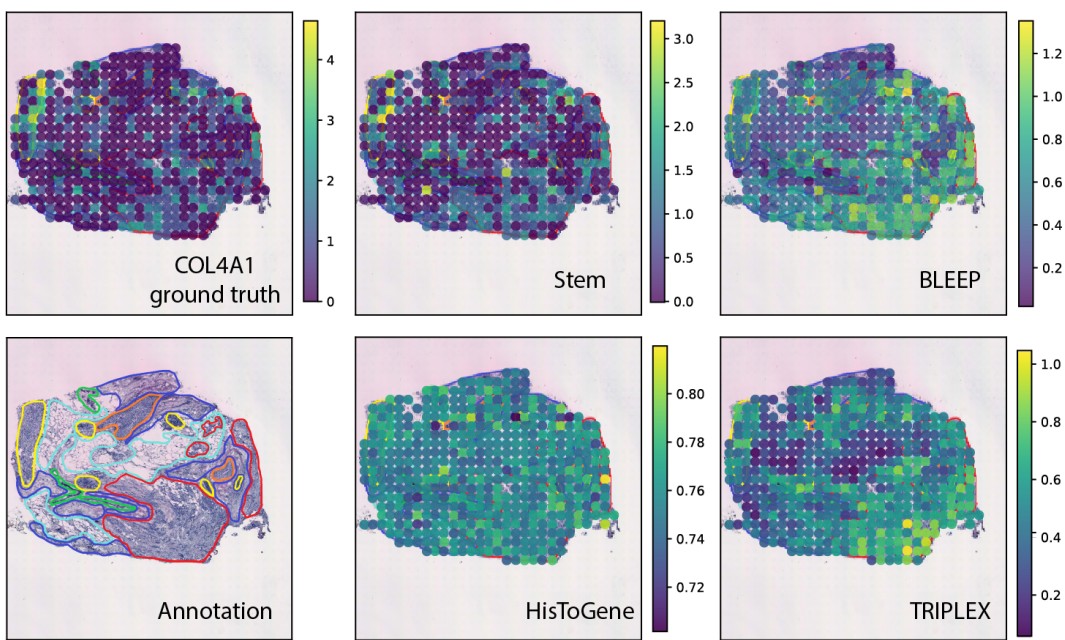

Figure 16: Visualization of marker gene COL4A1 predictions

