# OpenReview forum: "Diffusion Generative Modeling for Spatially Resolved Gene Expression Inference from Histology Images"
_ICLR.cc/2025/Conference — ICLR 2025 Poster_

### Official Review · Reviewer_8D3n · 2024-10-23

**Soundness:** 3
**Presentation:** 4
**Contribution:** 3
**Rating:** 6
**Confidence:** 5

**Summary:**

The authors proposed a novel computational tool that leverages a conditional diffusion generative model to enable in silico gene expression inference from H&E stained images. This tool can capture better heterogeneity and stochasticity of ST data and achieve SOTA on the gene expression prediction across various datasets with different metrics.

**Strengths:**

The authors considered a limitation I fully agreed with: gene expression cannot be gained from cellular morphology only. To solve this, proposed model learns a conditional distribution over the potentially associated gene expression profiles given the histology images, facilitating a one-to-many correspondence between the image and ST data.

**Weaknesses:**

I think the model should be tested with larger gene panel (maybe thousands of genes) to evaluate the performance limits compared with other methods. I also want to see the evaluation other than the datasets with super-resolution spots (for example, how about the performance under Slide-seq? Stereo-seq? or Even in situ technologies?). Does the image patch size change to match the size of the spots?

**Questions:**

1. I suggest testing the model with a larger gene panel (>1000) and compare with other methods.
2. I suggest testing on datasets generated by other platforms (Slide-seq, Stereo-seq, or in situ technologies).
3. For Figure 3, it is not enough to choose one sample with two marker gene. I would like to see more visualization results with more marker genes across different cell types. More importantly, compare these visualization results with existing methods. I would like to see the difference about cell type identification.
4. Since the authors used a transformed version of original histology images, I want to know how they transformed images.
5. How many genes did authors used for calculating MSE and MAE?

---

> ### Author Response · Authors · 2024-11-22
> **Response to Reviewer 8D3n (Part I)**
>
> We thank the reviewer for all the supportive and constructive comments. We are glad that the reviewer fully agrees with our motivation and finds our proposed algorithm novel. We provide a detailed response to your comments below.
>
> **Weakness 1 & Question 1 (Large Gene Panel)**
> > I think the model should be tested with larger gene panel (maybe thousands of genes) to evaluate the performance limits compared with other methods.
> > I suggest testing the model with a larger gene panel (>1000) and compare with other methods.
>
> Thank you for pointing out this meaningful direction to further test the robustness of Stem. To evaluate this, we perform an additional ablation experiment on a DEG gene set of size 1000 on HER2ST. We include the result in the paragraph **"Large gene sets"** in **Appendix B** of the updated manuscript. We can see from Table 6 that Stem still achieves the best performance in this setting and excels in most of the evaluation metrics. This result shows that Stem is scalable and maintains a consistently good performance when scaling to large gene set predictions. We kindly refer the reviewer to this paragraph for a more detailed discussion of the experiment.
>
> **Weakness 2 & Question 2 (Other platforms)**
> > I also want to see the evaluation other than the datasets with super-resolution spots (for example, how about the performance under Slide-seq? Stereo-seq? or Even in situ technologies?). Does the image patch size change to match the size of the spots?
> > I suggest testing on datasets generated by other platforms (Slide-seq, Stereo-seq, or in situ technologies).
>
> Thank you for the good suggestion! We believe that our proposed framework can be adapted to ST data produced by other platforms with super-resolution spot size, given that several technical conditions are met. We now discuss these conditions below.
>
> One key algorithmic component in **Stem** is the use of off-the-shelf pathology foundation models as image patch encoders. To adapt to super-resolution spot size, we would need a capable foundation model to encode the corresponding image patch (which has a much smaller diameter compared with the setting of Visium/ST platform mainly considered in this work). To the best of our knowledge, most existing pathology foundations models are only capable of encoding H&E stained histology image patches with a diameter larger than $40\mu m$, and super-resolution spot histology image patches do not qualify due to having a significantly smaller patch size. The lack of a super-resolution pathology foundation model is one key obstacle that prevents us from extending to platforms such as Slide-seq and Stereo-seq.
>
> A potential fix is to resize the super-resolution image patch to match the input requirement of existing foundation models. This requires the training data to have a raw high resolution since the image patch can't afford to lose too much resolution during the resizing. However, public datasets using Slide-seq or Stereo-seq with raw high-resolution H&E stained histology images are also very scarce and potentially hard to acquire.
>
> Due to the limited time during the rebuttal period, we weren't able to find available super-resolution foundation models or high-quality public datasets using Slide-seq or Stereo-seq that come with raw high-resolution histology images. Therefore, we weren't able to perform any numerical experiments for ST data with super-resolution spot size. If the reviewer happens to know some foundation models or datasets that suit the needs mentioned above, we would love to evaluate __Stem__ using them to further demonstrate the robustness of our proposed algorithm.

---

> ### Author Response · Authors · 2024-11-22
> **Response to Reviewer 8D3n (Part II)**
>
> **Question 3 (More Marker Genes)**
> > For Figure 3, it is not enough to choose one sample with two marker genes. I would like to see more visualization results with more marker genes across different cell types. More importantly, compare these visualization results with existing methods. I would like to see the difference about cell type identification.
>
> Thank you for this excellent suggestion! We appreciate the idea of visualizing more marker genes from other cell types. We include additional results in the **"Visualization of marker genes in HER2ST"** section in **Appendix E** of the updated manuscript. We visualize multiple marker genes for other cell types/tissue regions identified in the original paper for this HER2ST dataset [Fig.2 in Alma et al. 2021], such as CCL19 and TRAC for immune cells (including APC, B-cell, T-cell), IGHA1 for B/plasma cells, GPX3 for adipose tissue, RAB11FIP1 for immune rich in situ cancer, COL4A1 for mixture region of cancer and connective tissue. Again, as is clear from the plots, Stem consistently outperforms all other methods in generating gene expression value predictions with a better match to the ground truth gene expression level, in terms of both value scale and spatial patterns. We kindly refer the reviewer to this section for more details of the experiments.
>
> **Question 4 (Image Transformation)**
> > Since the authors used a transformed version of original histology images, I want to know how they transformed images.
>
> We use the following 7 transformations for image augmentation: horizontal flip, vertical flip, 90-degree rotation, 180-degree rotation, 270-degree rotation, transpose, and transverse. To better clarify this, we have included more explanations in the paragraph **"Image Augmentation"** in **Sec 5.3** of the updated manuscript. Thank you for helping us improve the paper's clarity!
>
> **Question 5 (MSE & MAE Computation)**
> > How many genes did authors used for calculating MSE and MAE?
>
> MSE and MAE are calculated using all the genes in the selected gene set. Thus the number of genes used varies between experiments. We have added more explanations regarding the computations of these metrics in the paragraph **"Evaluation metrics"** in the updated manuscript. Thank you again for the help in improving paper clarity!
>
> ---
> We hope the above responses could answer your questions. If you have any other questions, feel free to engage in the discussion and we would love to answer them. Thanks again for the supportive comments and all the efforts to improve this paper!
>
> ---
>
> Reference:
>
> Andersson, Alma, et al. Spatial deconvolution of HER2-positive breast cancer delineates tumor-associated cell type interactions (2021).
>
> Jaume, Guillaume, et al. Hest-1k: A dataset for spatial transcriptomics and histology image analysis (2024).

---

> ### Author Response · Authors · 2024-11-25
> **Sincerely Awaiting Your Feedback**
>
> Dear Reviewer 8D3n,
>
> I hope this message finds you well. As the rebuttal deadline is quickly approaching, we would greatly appreciate your valuable feedback to ensure a comprehensive revision. Your additional feedback would be immensely valuable in helping us ensure the highest quality of this work and address any remaining concerns you may have. We are looking forward to your response.
>
> Best regards,
> Authors

---

### Official Review · Reviewer_uQTZ · 2024-11-01

**Soundness:** 3
**Presentation:** 3
**Contribution:** 3
**Rating:** 6
**Confidence:** 3

**Summary:**

**Motivation:** Infer spatial resolved gene expression only from histology images
**Technical contribution:** A diffusion model based approach is proposed to predict spatial gene expression from histology images
**Strengths:** The paper is clearly written and the results support the story. The method is easy to understand and effective.
**Weakness:** The novelty of proposed method (clever application vs. novel technical contribution) is not very clear. Analysis can be expanded to more datasets.

**Justification of score:** Overall, the paper proposes a simple and effective method, which is above acceptance threshold. I would be happy to revise my score pending clarifications on my questions from the authors.

**Strengths:**

**Clear grounding of literature:** The introduction and related works sections are comprehensive and ground the problem well. In addition to working on better predicting gene expression, the authors also highlight problems with current simplistic evaluation framework based on Pearson correlation. They also propose new measures of correctness. Overall, the paper is clearly written.

**Comparison with recent baselines:** The authors benchmarked their model against recent baselines, such as BLEEP.

**Weaknesses:**

**Additional evaluation**
While the authors do demonstrate their method on two datasets and organs, there is a large amount of spatial transcriptomics data available publicly [1]. I encourage the authors to try it out and see if their method works equally well for different organs, cancer vs. non-cancer, and/ or different species.

**Using pre-trained encoders for gene expression**
The authors seem to train from scratch their gene value encoder. Can they comment on why do they do this when numerous gene expression encoder foundation models are available, such as scGPT [2]? While these encoders have been trained on single cell data, they have shown promising results for bulk expression, hence showing their versatility.

**Limited novelty**
The proposed method seems like a clever application of conditional diffusion models to the problem. Can the authors further comment on the novelty of their method and how is it different compared to the existing literature?



---
References
[1] Jaume, Guillaume, et al. "Hest-1k: A dataset for spatial transcriptomics and histology image analysis." arXiv preprint arXiv:2406.16192 (2024).
[2] Cui, Haotian, et al. "scGPT: toward building a foundation model for single-cell multi-omics using generative AI." Nature Methods (2024): 1-11.

**Questions:**

- Was any batch correction applied to the spatial transcriptomics data?
- Since many metrics are presented in the table, it might be easier to interpret them if authors have conventional up and down arrows next to the metric name.
- What was the rationale behind trying only UNI and CONCH, when larger models exist, such as Virchow [1]?

---
References
[1] Vorontsov, Eugene, et al. "A foundation model for clinical-grade computational pathology and rare cancers detection." Nature medicine (2024): 1-12.

---

> ### Author Response · Authors · 2024-11-22
> **Response to Reviewer uQTZ (Part I)**
>
> We thank the reviewer for all the supportive and constructive comments. We are glad that the reviewer finds our paper clearly written with comprehensive numerical experiments and our proposed approach novel and effective. We provided detailed responses to your comments below.
>
> **Weakness 1 (Robustness to datasets)**
> > While the authors do demonstrate their method on two datasets and organs, there is a large amount of spatial transcriptomics data available publicly [1]. I encourage the authors to try it out and see if their method works equally well for different organs, cancer vs. non-cancer, and/ or different species.
>
> Thank you for the great suggestion! To further demonstrate the robustness of Stem to datasets, we additionally perform experiments on two datasets, with one being a human prostate cancer Visium dataset and the other one being a healthy mouse brain Visium dataset. These two datasets are different in organs, species, and health conditions from the two datasets presented in the main text. We include the numerical results in **Appendix D**, where we present both metric values and the plots for gene variation curves. Based on the results, we conclude that Stem is very robust to dataset choices. On both datasets, it consistently outperforms other methods in terms of better evaluation metric values and better gene variation curves (which are closer to the ground truth). We kindly refer the reviewer to **Appendix D** for a more detailed description of the two new datasets and a more thorough discussion of the numerical results.
>
> **Weakness 2 (Off-the-self Gene Encoder)**
> > The authors seem to train from scratch their gene value encoder. Can they comment on why do they do this when numerous gene expression encoder foundation models are available, such as scGPT [2]? While these encoders have been trained on single cell data, they have shown promising results for bulk expression, hence showing their versatility.
>
> Thank you for bringing up this interesting idea. While existing gene expression encoder foundation models such as scGPT are powerful, unfortunately, they are not directly applicable to our algorithm due to the fact that we need to encode both clean and noisy gene expression profiles during diffusion model training. When generating samples from diffusion models, we start with a gene expression that is pure Gaussian noise and then iteratively denoise the gene vector for sample refinement. Eventually, this procedure produces a valid gene expression vector. This means that our neural network, especially the gene encoder, should be capable of encoding gene expression profiles corrupted with various levels of Gaussian noise into meaningful latent representations. Presumably, this is a task that gene expression foundation models can't directly perform, since most foundation models like scGPT can only operate on clean data during both the training and inference phases. Therefore, we choose to train our own encoder and do not resort to existing gene foundation models.

---

> ### Author Response · Authors · 2024-11-22
> **Response to Reviewer uQTZ (Part II)**
>
> **Weakness 3 (Novelty)**
> > The proposed method seems like a clever application of conditional diffusion models to the problem. Can the authors further comment on the novelty of their method and how is it different compared to the existing literature?
>
> Thank you for allowing us to further clarify the novelty of Stem compared with existing methods. Stem is indeed a novel application of conditional diffusion models to the Spatial Transcriptomics prediction problem. We have briefly summarized the existing approaches on this task and compared Stem with them in the paragraph **"Machine Learning Prediction of Gene Expression from Histology Images"** in **Sec 2**, and we will provide a more detailed explanation below.
>
> 1. Compared with the ST-prediction literature, Stem is **the first generative modeling algorithm** that models a gene expression probability distribution to better capture spatial heterogeneity. In contrast, existing works such as BLEEP, TRIPLEX, or HisToGene are **regression-based** approaches that produce deterministic predictions that suffer from a lack of spatial diversity, which leads to a low accuracy on downstream tasks such as unsupervised tissue structure annotations (Fig.3 in the main text). Stem also differs from existing approaches in that it distills knowledge from existing foundation models by using off-the-shelf histopathology foundation models as patch encoders, while previous approaches haven't taken the opportunity to explore this design choice.
>
> 2. Compared with the diffusion model (DM) literature, Stem successfully expands the application domain of multimodal diffusion models and includes Histology-to-ST as a new task. Stem also contributes to the literature of DMs by experimenting with new ideas in conditioning mechanisms, which is a crucial algorithmic component in conditional DMs. Stem successfully extends the modulation approach proposed in DiT for enforcing simple class-label conditioning to sophisticated histology image conditioning, the latter usually considered to be a much more difficult task and requires a more complex conditioning mechanism design than a simple modulation module. Stem shows that the modulation mechanism has more potential than people used to believe, which is a potentially important message to other multimodal applications such as text-to-image and text-to-3D generation.
>
>
> **Question 1 (Batch Correction)**
> > Was any batch correction applied to the spatial transcriptomics data?
>
> In our experiments, we do not apply any batch correction algorithm to the data. We follow the preprocessing protocol used in the Hest-1k dataset benchmark [Jaume et al. 2024], which does not contain any batch correction procedure. Based on the performance of __Stem__, we observed that it is robust to batch effect and technical variations at a slide level (i.e. in kidney dataset, almost all ST slides are from different patients).
>
> **Question 2 (Arrows)**
> > Since many metrics are presented in the table, it might be easier to interpret them if authors have conventional up and down arrows next to the metric name.
>
> Thank you for the thoughtful suggestions on making the paper more readable! We have added up and down arrows next to the metric names in all the tables of the updated manuscript to help readers better comprehend them.

---

> ### Author Response · Authors · 2024-11-22
> **Response to Reviewer uQTZ (Part III)**
>
> **Question 3 (Large Pathology Foundation Model)**
> > What was the rationale behind trying only UNI and CONCH, when larger models exist, such as Virchow [1]?
>
> Thank you for the great question! Our initial motivation behind using UNI and CONCH is to test the robustness of __Stem__ regarding different types of patch embeddings generated by foundation models. While UNI and CONCH are both powerful foundation models, they are trained with different self-supervised learning algorithms. UNI is a vision-only model pre-trained with DINOv2 objective [Oquab et al. 2023] and CONCH is a vision-language model pre-trained with a contrastive image captioning objective [Yu et al. 2022]. In principle, this suggests that UNI and CONCH are capable of capturing distinct features from the same image patch. Later, we found that not only __Stem__ is robust to different types of foundation model embeddings, UNI and CONCH also seemingly provide complementary information for histology image patches since leveraging combined patch embeddings from UNI and CONCH produces the best numerical performance (See **Sec 5.3**, paragraph **"Choice of foundation model"**).
>
> To make the experiments more comprehensive and convincing, we include additional results for __Stem__ using Virchow-2 and H-Optimus-0, which are trained in a similar way as UNI (based on DINOv2 with ViT backbone) but of a much larger scale. We present the additional results in the paragraph **"Large Pathology Foundation Models"** in Appendix B of the updated manuscript. Based on the results in Table 8, we reach the conclusion that Stem does not necessarily perform better with larger pathology foundation models, since UNI surpasses Virchow-2 and H-Optimus-0 on many metrics. However, using UNI/Virchow-2/H-Optimus-0 alone still leads to a subpar performance compared with using the combination of CONCH and UNI. This result accords with our intuition mentioned above, and it further justifies our choice of CONCH + UNI as the default, best-performing setting for Stem. We kindly refer the reviewer to this paragraph for a more detailed discussion.
>
> ---
>
> We hope the above responses could answer your questions. If you have any other questions, feel free to engage in the discussion and we would love to answer them. Thanks again for the supportive comments and all the efforts to improve this paper!
>
> ---
>
> Reference:
>
> Oquab, Maxime, et al. Dinov2: Learning robust visual features without supervision (2023)
>
> Yu, Jiahui, et al. Coca: Contrastive captioners are image-text foundation models (2022)
>
> Jaume, Guillaume, et al. Hest-1k: A dataset for spatial transcriptomics and histology image analysis (2024)

---

> > ### Comment · Reviewer_uQTZ · 2024-11-25
> >
> > Thank you for answering my questions and presenting additional results.

---

> > > ### Author Response · Authors · 2024-11-25
> > >
> > > Dear Reviewer uQTZ,
> > >
> > > Thank you for your response! We would like to know if there‘s any additional concern that you still have regarding this work, and we would be eager to address them thoroughly. Is there anything else we can do to earn your re-evaluation and improved support for this work?
> > >
> > > Best regards,
> > > Authors

---

### Official Review · Reviewer_VznU · 2024-11-03

**Soundness:** 3
**Presentation:** 3
**Contribution:** 3
**Rating:** 8
**Confidence:** 4

**Summary:**

The authors propose STEM, which is a conditional diffusion generative model-based ST prediction framework. On two public datasets, the authors show that STEM outperforms the state-of-the-art ST prediction methods. The authors also show through a series of ablation experiments what factors are contributing to good performance of their model.

**Strengths:**

I think STEM adds a valid contribution to a large suite of ST-prediction methods, which are predominantly regression-based approaches (except for BLEEP, which relies on the embedding neighborhood in the training set). The application of generative model is novel, the strength of which is clearly shown in the great performance. I also appreciate that the authors invested efforts into several ablation study factors and carefully designed the experiments to mitigate potential bias (which I don't think BLEEP did well at all). Assuming that the codebase is provided, I think this can be integrated into pratictioner's toolbox.

**Weaknesses:**

I think there are several weaknesses that the authors can address to make it an even better contribution to the field.

- For reproducibility, the authors NEED to be more specific about hyperparameters used in STEM - I don't think any information was provided.
- This is a generative model, meaning multiple gene predictions are generated from the given histology image. While the authors say that these are averaged to yield the final expression, I would like to see an ablation or example of what each sample looks like and how they would affect all the performance metrics that we observe.
- While not required, I would like to see the authors also try with different histopatholgy foundation models, such as Virchow or H-Optimus, just for robustness
- I am slightly confused as to why TRIPLEX performance is really low. In their CVPR 2024 paper, they show that TRIPLEX outperforms HisToGene and BLEEP significantly. But it is really bad. Have authors used their model properly?
- To really show that it is indeed the *conditional generative model* part of STEM contributing to the increase PCC performance, the authors need to be careful in comparing to other SOTA baselines. BLEEP uses ResNet50, TRIPLEX uses Resnet18, whereas STEM uses UNI or CONCH. Could the difference simply be coming from difference in patch encoders?
- The authors (in line 78~80) argue that even same cell type might be in different cell types of differ in spatial locations causing different gene expression outcomes and that previous models cannot 'capture them. However, I am not sure if STEM can also address this shortcoming of the previous models, since it is also simply using patch encoder (albeit a powerful one) to summarize the image patch.
- When encoding gene count, the input is a scalar, passed through MLP? The authors need to elaborate more on what this MLP structure is, since it is confusing.
- In encoder ablation study, what does CONCH+UNI mean? Simple concatenation of both features?

**Questions:**

See weakness

---

> ### Author Response · Authors · 2024-11-22
> **Response to Reviewer VznU (Part I)**
>
> We thank the reviewer for the supportive and constructive comments. We are glad that the reviewer recognizes our work as a novel, solid contribution to the ST prediction and enjoy the comprehensive ablation study that we performed. We will soon make our codebase public so that the practitioners can use it in practice. We provide responses to each of your comments below.
>
> **Weakness 1 (Reproducibility)**
> > For reproducibility, the authors NEED to be more specific about hyperparameters used in STEM - I don't think any information was provided.
> >
> Thank you for bringing up this issue. We agree with the reviewer, and to improve the reproducibility of the paper, we include a detailed description of the neural architecture design and training hyperparameters in **Appendix C** in the updated manuscript. We will also include all the hyperparameter configurations for each experiment in the codebase for release.
>
> **Weakness 2 (Generated Samples and Sample Statistics)**
> > This is a generative model, meaning multiple gene predictions are generated from the given histology image. While the authors say that these are averaged to yield the final expression, I would like to see an ablation or example of what each sample looks like and how they would affect all the performance metrics that we observe.
>
> Thank you for the great question! To answer this question, we conduct a new ablation study on the influence of different sample statistics on the numerical performances of __Stem__. We include the results in paragraph **"Generated Samples and Sample Statistics"** in **Appendix B** of the updated manuscript. We visualize the generated samples, and predictions using different sample statistics and ground truth expression values on randomly selected gene pairs. Other than sample mean, we also experiment with sample median and sample mode as two alternative approaches to generate predictions from samples. Results for the performance of __Stem__ under different sample statistics are presented in Table 7. While in general sample means prevails in most metrics, other statistics can produce a more accurate prediction under certain evaluation metrics as well. We kindly refer the reviewer to this paragraph for more details.
>
>
> **Weakness 3 (More Pathology Foundation Model)**
> > While not required, I would like to see the authors also try with different histopathology foundation models, such as Virchow or H-Optimus, just for robustness
>
> Thank you for pointing out this interesting direction to test algorithm robustness. We perform an additional ablation study focusing on using large pathology foundation models as histology image patch encoders, and include the new results in the paragraph **"Large Pathology Foundation Models"** in **Appendix B**. We benchmark the performance of Stem using Virchow-2 and H-Optimus-0 on the Kidney Visium dataset. As is clear from the metrics in Table 8, Stem is robust to the choice of foundation models and works equally well with Virchow-2 and H-Optimus-0, although using a combination of UNI and CONCH still gives the best numerical performance if judging from the metric values. We kindly refer the reviewer to this paragraph for a more detailed discussion on the effects of model size of pathology foundation models.
>
> **Weakness 4 (TRIPLEX Performance)**
> > I am slightly confused as to why TRIPLEX performance is really low. In their CVPR 2024 paper, they show that TRIPLEX outperforms HisToGene and BLEEP significantly. But it is really bad. Have authors used their model properly?
>
> Thank you for noticing this issue! Only after the ICLR submission deadline had passed, we discover that there were certain hyperparameters and data preprocessing steps (specific to their settings) hardcoded in the original code implementation of TRIPLEX, which was not explicitly mentioned or instructed by the authors of TRIPLEX. This led to an improper and sub-optimal performance of TRIPLEX when being adapted to our settings and new datasets. We have now fixed the implementation errors and correctly evaluated TRIPLEX on all the experiments. We have updated Table 1, and Table 2 in the paper with the correct values. We have also updated plots for gene variation curves in **Appendix A** (Fig 4 to Fig 7) to include a visualization of the curves produced by TRIPLEX. With the updated performance of TRIPLEX, Stem still achieves SOTA performance among all the existing methods, with an especially large lead in the spatial heterogeneity metric RVD.

---

> ### Author Response · Authors · 2024-11-22
> **Response to Reviewer VznU (Part II)**
>
> **Weakness 5 (Influence of Patch Encoder)**
> > To really show that it is indeed the conditional generative model part of STEM contributing to the increase PCC performance, the authors need to be careful in comparing to other SOTA baselines. BLEEP uses ResNet50, TRIPLEX uses Resnet18, whereas STEM uses UNI or CONCH. Could the difference simply be coming from difference in patch encoders?
>
> Thank you for the good question! To indeed show that the formulation of conditional diffusion models gives Stem a nontrivial improvement in numerical performances, we perform an additional ablation study focusing on benchmarking the representation power of different pre-trained image patch encoders adopted in the literature, such as ResNet18 (used in TRIPLEX), UNI, and CONCH. We include the new results in the paragraph **"Power of Histology Image Patch Encoder"** in **Appendix B**. We build a simple yet effective protocol to generate gene expression predictions using these pre-trained encoders, following a similar procedure used in BLEEP. We list the results in Table 9 and compare them with the performance of Stem. Surprisingly, the seemingly strongest image encoder CONCH + UNI does not achieve the best value in any metric, while ResNet18 used in TRIPLEX tops two PCC values. Moreover, Stem outperforms all these benchmarks by a large margin in any metric, including the CONCH + UNI benchmark which shares the same image patch encoder as Stem. This suggests that other algorithm components such as conditional diffusion models have essential contributions to the improved performance of Stem over existing approaches. We kindly refer the reviewer to this paragraph for a more detailed discussion of this ablation experiment.
>
> **Weakness 6 (Spatial Heterogeneity)**
> > The authors (in line 78~80) argue that even same cell type might be in different cell types of differ in spatial locations causing different gene expression outcomes and that previous models cannot 'capture them. However, I am not sure if STEM can also address this shortcoming of the previous models, since it is also simply using patch encoder (albeit a powerful one) to summarize the image patch.
>
> Thank for you bringing up this question and allowing us to further clarify this aspect. In lines 78-80, we described two scenarios where previous methods are likely to fail. We will explain in detail below how Stem can outperform previous methods in addressing these issues even by using only patch encoders.
>
> 1. First scenario: two cells of near identical cell morphology (i.e. similar histology image patch) with different gene expression profiles at different locations. In this case, the underlying ground truth distribution associated with this cell morphology/histology image patch should be a mixture of at least two distinct points. Stem has a natural advantage in this situation since it can capture exactly this mixture distribution during training due to being a generative approach. On the contrary, existing approaches are forced to learn a deterministic prediction, which causes a loss in the learned information and hence a sub-optimal performance.
>
> 2. Second scenario: two cells with different cell morphology and distinct gene expression profiles at different locations. In this case, Stem can also outperform existing approaches, judging from the numerical results in terms of Pearson correlations, MSE, MAE, RVD, and heatmap visualizations of gene expressions. Moreover, we noticed that Stem is significantly more capable of capturing spatial heterogeneity and diversity than other approaches. This can be seen by looking at the figures of gene variation curves (see Fig 4 to Fig 7 in **Appendix A**), where it's clear that Stem produces gene expression predictions with the best match to the ground truth variation curve. It is also directly reflected in Stem's significantly lower RVD value compared with other approaches, as is presented in Table 1 and Table 2.

---

> ### Author Response · Authors · 2024-11-22
> **Response to Reviewer VznU (Part III)**
>
> **Weakness 7 (Gene Embedding)**
> > When encoding gene count, the input is a scalar, passed through MLP? The authors need to elaborate more on what this MLP structure is, since it is confusing.
>
> One gene count is indeed a 1-dimensional scalar value, and we transformed it into a latent vector with dimension $D$ using an MLP. We choose the first layer in the MLP to be a linear layer with input dimension 1 and output dimension $D$ so that the scalar gene count value becomes a vector. We explain this procedure in detail in **Appendix C** of the updated manuscripts. Thank you for helping us improve the paper's clarity!
>
> **Weakness 8 (CONCH + UNI)**
> > In encoder ablation study, what does CONCH+UNI mean? Simple concatenation of both features?
>
> Yes, CONCH + UNI refers to the procedure of extracting patch embedding using UNI and CONCH respectively, and then concatenating the two embeddings. We add more descriptions of this procedure in the updated manuscript. Thank you again for helping us with paper clarity!
>
> ---
> We hope the above responses could answer your questions. If you have any other questions, feel free to engage in the discussion and we would love to answer them. Thanks again for the supportive comments and the efforts to improve this paper!

---

> ### Public Comment · ~Yuhan_Wang11 · 2024-11-23
> **Gene Embedding Construction**
>
> Thank for your interesting work. I am very interested in this field, but I have a little doubt. I understand that your gene encoder maps 1-dimensional scalar values to dimension D, However, Appendix C does not mention how to map decoder back to 1-dimensional scalar value for quantification. My understanding is that you also mapped the ground truth of gene expression values to the high dimension D through the pretrained 2-layer mlp and calculated mertics with the predictions of diffusion model. But is this quantification inconsistent with existing work such as Bleep, and will it lead to unfair comparisons. I would appreciate it if you could explain my doubt.

---

> ### Author Response · Authors · 2024-11-23
> **More explanation on relationship between neural network output and diffusion model**
>
> Thank you for the question. Please allow us to further explain. First of all, our algorithm outputs **valid gene expression vectors** instead of their high-dimension embeddings, and we compare the generated gene expression vector with the ground truth gene expression vector, which is **the same evaluation protocol** considered in existing works such as BLEEP and is a fair comparison.
>
> Secondly, our model uses a simple decoder to map the latent embedding vector $D$ to some low-dimensional object that is required by the diffusion model to generate the scalar count value. This procedure is mentioned in Lines 332-333, which we did not expand on details since we inherit the same practice as DiT. For the self-consistency of the content and to further improve the paper's clarity, we now add the description of this decoder in **Appendix C**.
>
> We now provide below a clarification on the neural network output, and how it relates to the generation of gene expression vectors through diffusion models.
>
> Our neural network, denoted as $s_{\theta}(x_t, t)$, takes an input $x_t \in \mathbb{R}^{C}$ and $t \in {0, \dots, T}$, where $C$ is the dimension of the gene expression vector, and it eventually outputs a vector $\mu_{\theta}(x_t, t) \in \mathbb{R}^{C}$. At each step of the diffusion model sampling, the prediction for the next step value $x_{t-1}$ is sampled from a Gaussian distribution with mean $\mu_{\theta}(x_t, t)$ and a pre-determined $t$-dependent variance matrix. This procedure is repeated until $t = 0$, and the generated $x_0$ is **a valid gene expression vector**, with distribution approximately the same as ground truth values. More information on diffusion models is presented in paragraph **Diffusion Model** in **Sec 3** and paragraph **Diffusion Generative Modeling of Gene Expressions** in **Sec 4**.
>
> $\mu_{\theta}(x_t, t)$ is computed in the following procedure. With our gene encoder, $x_t$ is first transformed into a sequence of $C$ latent embedding vectors, each of dimension $D$. Then this sequence of latent embeddings is jointly passed through DiT transformer blocks with the output being the same shape, thus an element in $\mathbb{R}^{C \times D}$. Finally, the output of the DiT blocks is fed to the simple decoder and the decoded output is in $\mathbb{R}^{C \times 1}$, which is then used to compute $\mu_{\theta}(x_t, t)$ following equation in Line 289.
>
> I hope the above explanation can answer your question and successfully explain why our evaluation protocol is fair. We are happy to clarify more if needed. Thank you again for engaging in the discussion.

---

> ### Author Response · Authors · 2024-11-25
> **Sincerely Awaiting Your Feedback**
>
> Dear Reviewer VznU,
>
> I hope this message finds you well. As the rebuttal deadline is quickly approaching, we would greatly appreciate your valuable feedback to ensure a comprehensive revision. Your additional feedback would be immensely valuable in helping us ensure the highest quality of this work and address any remaining concerns you may have. We are looking forward to your response.
>
> Best regards,
> Authors

---

> > ### Comment · Reviewer_VznU · 2024-11-26
> >
> > Dear authors,
> >
> > Thank you for addressing all my questions with thorough details - With additional analyses, I think the study is much stronger now than before. I will adjust my score accordingly

---

> > > ### Author Response · Authors · 2024-11-26
> > >
> > > Dear Reviewer VznU,
> > >
> > > We deeply appreciate your response and your support for our work! Your specific suggestions on additional analyses have been immensely valuable in helping us improve the quality of this manuscript.
> > >
> > > Thank you again for your valuable feedback and support.
> > >
> > > Best regards,
> > > Authors

---

### Author Response · Authors · 2024-11-22
**Common Reply and Updated Manuscript**

We thank all reviewers for their supportive comments and constructive suggestions for this paper. We deeply appreciate reviewers' recognition of the importance of generative modeling in the task of predicting spatially resolved gene expression. We are glad to see that all three reviewers agree that our proposed method has a solid contribution, with a concrete motivation and clear presentation. We have addressed all the questions and comments raised by all three reviewers and further improved the quality of the paper. We are very grateful for all the precious suggestions and efforts made to better this work.

We have revised our paper in response to the comments and updated it in the portal. To make it easier for the reviewers to track the updates, **the changes made in the main text are highlighted in purple**. We do not highlight changes in the Appendix due to the large amount of new materials added. We summarize the major changes in the items below.


**Main text**
- **Line 340**, we clarified the number of genes used in the computation of MAE and MSE, as suggested by reviewer 8D3n.
- **Line 485-486**, we added the explanation for CONCH + UNI, as suggested by reviewer VznU
- **Line 495-498**, we added a detailed explanation for image transformations used in augmentation, as suggested by reviewer 8D3n.

**Appendix**
- **Appendix A**, we updated the plots for gene variation curves on the Kidney Visium and HER2ST dataset to include all methods for comparison
- **Appendix B**, we included details for four new ablation experiments, with the following new paragraphs as jointly suggested by all three reviewers,
    - __Large gene sets__, focus on __Stem__'s performance on a gene panel of size 1000, as suggested by reviewer 8D3n
    - __Generated Samples and Sample Statistics__, focus on how different samples contribute to the algorithm performance, as suggested by reviewer VznU
    - __Large Pathology Foundation Models__, focus on evaluating **Stem** with large-scale pathology foundation models, as suggested by reviewer VznU and uQTZ
    - __Power of Histology Image Patch Encoder__, focus on investigating the contribution of image patch encoder to the overall performance of __Stem__, as suggested by reviewer VznU
- **Appendix C**, we added a detailed explanation for neural network architecture and training procedures to improve paper reproducibility, as suggested by reviewer VznU
- **Appendix D**, we included numerical results on two new datasets with different organs, species, and health conditions from the ones considered in the main text, as suggested by reviewer uQTZ
- **Appendix E**, we added additional visualization for other cell-type-specific marker genes predictions, as suggested by reviewer 8D3n

---

### Meta-Review · Area_Chair_EPGq · 2024-12-21

**Metareview:**

The paper introduces STEM, a conditional diffusion generative model designed to predict spatial gene expression from histology images, achieving state-of-the-art performance across multiple datasets and evaluation metrics. The model effectively captures biological heterogeneity by learning a one-to-many mapping between histology images and spatial transcriptomics data, addressing key limitations of existing approaches. While reviewers acknowledged the paper's strengths, they highlighted areas for further improvement, including broader evaluation on datasets such as Slide-seq and Stereo-seq, testing with larger gene panels, and providing more comprehensive visualizations comparing results with existing methods. Although many of these concerns were adequately addressed by the authors, the evaluation on Slide-seq and Stereo-seq datasets remains unaddressed due to the lack of a super-resolution pathology foundation model . Overall, it is well-motivated, clearly written, and demonstrates improvements over existing baselines with comprehensive experiments and ablation studies, leading to a consensus in acceptance among all reviewers.

**Additional Comments On Reviewer Discussion:**

The concerns are well addressed during the discussion phase.

---

### Decision · Program_Chairs · 2025-01-22

Accept (Poster)